# Real-time prediction of COVID-19 related mortality using electronic health records

Patrick Schwab [1✉], Arash Mehrjou [2,3], Sonali Parbhoo[4], Leo Anthony Celi[5,6], Jürgen Hetzel[7,8], Markus Hofer[8], Bernhard Schölkopf[2,3] & Stefan Bauer [2,9]

Coronavirus disease 2019 (COVID-19) is a respiratory disease with rapid human-to-human transmission caused by the severe acute respiratory syndrome coronavirus 2 (SARS-CoV-2). Due to the exponential growth of infections, identifying patients with the highest mortality risk early is critical to enable effective intervention and prioritisation of care. Here, we present the COVID-19 early warning system (CovEWS), a risk scoring system for assessing COVID-19 related mortality risk that we developed using data amounting to a total of over 2863 years of observation time from a cohort of 66 430 patients seen at over 69 healthcare institutions. On an external cohort of 5005 patients, CovEWS predicts mortality from 78.8% (95% confidence interval [CI]: 76.0, 84.7%) to 69.4% (95% CI: 57.6, 75.2%) specificity at sensitivities greater than 95% between, respectively, 1 and 192 h prior to mortality events. CovEWS could enable earlier intervention, and may therefore help in preventing or mitigating COVID-19 related mortality.

[1] F. Hoffmann-La Roche Ltd, Basel, Switzerland. [2] Max Planck Institute for Intelligent Systems, Tübingen, Germany. [3] ETH Zurich, Zurich, Switzerland. [4] John A. Paulson School of Engineering and Applied Sciences, Harvard University, Cambridge, USA. [5] Department of Medicine, Beth Israel Deaconess Medical Center, Harvard Medical School, Boston, USA. [6] MIT Critical Data, Laboratory for Computational Physiology, Institute for Medical Engineering and Science, Harvard-MIT Health Sciences and Technology, Cambridge, USA. [7] Department of Medical Oncology and Pneumology, University Hospital of Tübingen, Tübingen, Germany. [8] Department of Pneumology, Kantonsspital Winterthur, Winterthur, Switzerland. [9] CIFAR Azrieli Global Scholar, Toronto, Canada. ✉email: patrick.schwab@icloud.com

The coronavirus disease 2019 (COVID-19) pandemic has recently emerged as a major and urgent threat to healthcare systems worldwide. Since early reports of its outbreak in China in December 2019, the number of global cases has risen to over 21 million known infections and resulted in over 750,000 deaths worldwide as of August 16, 2020[1]. Despite public health efforts aimed at improving testing[2], developing potential vaccines[3], and improving prevention strategies[4], the disease is placing a significant burden on healthcare systems and existing resources in many countries, particularly where its spread has not been mitigated. Efficient early detection of patients likely to develop critical illness is thus crucial to optimise the allocation of limited resources, and monitor overall disease progression[5,6]. The use of clinical predictive models from electronic health records (EHRs) can help reduce some of this burden and inform better decisions overall[7–10]. For instance, a model able to predict in advance which patients are at higher risk of mortality may help ensure resources are prioritised accordingly for these individuals. In addition, as more observational data are gathered, these models could be used both to discover new risk factors, as well as reveal interactions between existing factors, offering better insights and opportunities for appropriate intervention.

Several approaches have been proposed to determine potential risk factors that contribute to COVID-19 mortality. Some of these approaches identify demographics and inflammatory markers associated with increased mortality[11,12], but do not account for risk factors potentially changing over time. Moreover, many existing analyses are limited to a single source of data, often from a single hospital, for both learning a model and predicting a patient's risk which may limit the generalisability of these analyses[13]. Other more traditional measures of patient prognosis such as sequential organ failure assessment (SOFA) scores[14] are based on examining a fixed set of risk factors not specifically adapted to COVID-19; such measures fail to account for relevant changes in patient status outside these risk factors, and therefore often do not reach high levels of sensitivity and specificity in identifying high-risk patients. Due to these challenges, to date, there does not yet exist a COVID-19 risk score that (i) makes use of multiple, representative sources of data to account for patient heterogeneity, (ii) includes important short-term and long-term risk factors that have a significant impact on mortality risk, (iii) reacts in real time to potentially rapid changes in patient status, and (iv) is adapted to risk factors relevant to COVID-19.

To address these issues, we developed the COVID-19 early warning system (CovEWS), a risk assessment system for real-time prediction of COVID-19-related mortality that we trained on a large and representative sample of EHRs collected from more than 69 healthcare institutions using machine learning. In contrast to existing risk scores, CovEWS provides early warnings with clinically meaningful predictive performance up to 192 h prior to observed mortality events, hence enabling critical time to intervene to potentially prevent such events from occurring. Since CovEWS is automatically derived from patient EHRs, it updates in real time without any necessity for manual action to reflect changes in patient status, and accounts for a much larger number of risk factors correlated with COVID-19 mortality than existing risk scores. CovEWS was derived from the de-identified EHRs of 66,430 diverse COVID-19 patients, and is based on a time-varying neural Cox model that accounts for risk factors changing over time and potential non-linear interactions between risk factors and COVID-19-related mortality risk. While these extensions have been pursued in[15] and[16] separately, they have neither been considered in combination nor in the context of COVID-19 risk scoring using EHRs. We demonstrate experimentally that the predictive performance of CovEWS is superior to existing generic risk scores, such as SOFA[14], COVID-19

specific risk scores, such as the machine learning models from Yan et al.[17] and Liang et al.[18], and COVER_F[19], and a time-varying Cox model with linear interactions[20]. We additionally show that the gradient information of our differentiable CovEWS model can be used to quantify the influence of the input risk factors on the output score in real time. CovEWS may enable clinicians to identify high-risk patients at an early stage, and may, therefore, help improve patient outcomes through earlier intervention.

## Results

**COVID-19 early warning system (CovEWS).** CovEWS is a clinical mortality risk prediction system for COVID-19 positive patients to be used in a continuous manner in both inpatient and outpatient settings. CovEWS uses clinical risk factors from a patient's EHR to automatically calculate a mortality risk score between 0 and 100 that indicates the current risk percentile that this patient is in relative to the reference cohort (see "Model" for a mathematical definition of CovEWS). A CovEWS score of 90 indicates, for example, that the patient has a higher COVID-19 related mortality risk than 90% of COVID-19 positive patients in the reference cohort. An important property of CovEWS scores is that they always reflect the momentary risk of patients in their current states, and that they update instantaneously to reflect relevant, EHR-derived changes, which is a key differentiator of CovEWS compared to existing COVID-19 related mortality risk prediction systems that are not designed to take into account new, incoming clinical evidence. Figure 1 demonstrates the application of CovEWS to two contrasting patient timelines (a deteriorating patient that ultimately died and a patient that initially deteriorates but then recovers) by visualising a selected number of clinical risk factors, such as respiratory rate, oxygen saturation, and creatinine levels, alongside the corresponding momentary risk assessment output by CovEWS. As shown in Fig. 1, CovEWS additionally maintains a high degree of interpretability for clinicians by indicating the relative positive and negative influences of each clinical risk factor over time on the predicted risk score (see "Feature Importance"). The information conveyed by CovEWS can be used to quickly and objectively assess individual COVID-19 related mortality risk in order to prevent or mitigate mortality, and optimise prioritisation of scarce healthcare resources.

To develop CovEWS, we used EHR data from two federated networks of US and international healthcare organisations (HCOs), Optum (US) and TriNetX (US + international), that include de-identified EHRs containing data on demographics, clinical measurements, vital signs, lab tests and diagnoses of 47,384 and 5005 patients seen between March 21st and June 5th 2020 (11 weeks) and March 21st and June 25st 2020 (13 weeks), respectively. To demonstrate the generalisability of predictions made by CovEWS, we limited the training of CovEWS to a training cohort of 23,692 (50%) patients from the Optum cohort, used 9477 (20%) Optum patients for model selection, and evaluated CovEWS against both a held-out test cohort of 14,215 (30%) patients from the Optum cohort and a separate external test cohort consisting of the entire TriNetX cohort of 5005 (100%) patients (Table 1, stratification details in "Stratification"). In addition, we collected supplementary EHR data on new patients diagnosed with COVID-19 between June 6th to July 13th 2020 (5 weeks) from Optum—the Optum future cohort (14,041 patients)—after CovEWS had been trained to demonstrate the robustness of CovEWS under rapidly changing treatment regimes and other temporal effects. We note that during this period, the RECOVERY Collaborative Group reported results of randomised clinical trials demonstrating the lack of efficacy of hydroxychloroquine[21] and the efficacy of dexamethasone[22] in COVID-19

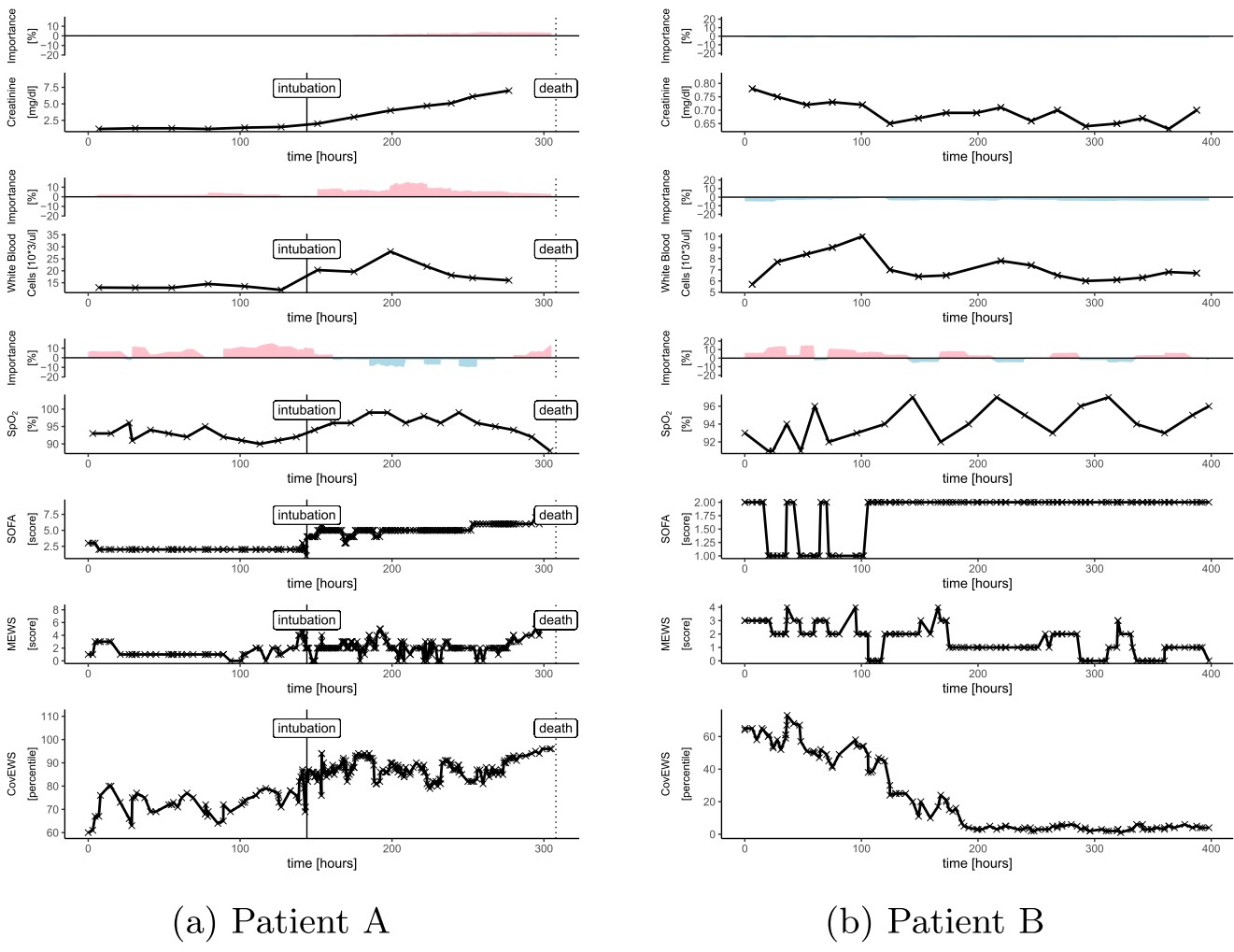

(a) Patient A

(b) Patient B

**Fig. 1 A selected number of clinical risk factors, and corresponding SOFA, modified early warning score (MEWS) and CovEWS scores for two contrasting patient timelines.** Positive (red) and negative (blue) importance contributions (coloured areas above the clinical time series, see Section "Feature Importance") indicate to what degree the risk factor at that time point contributed to increasing or decreasing to the mortality risk predicted by CovEWS, respectively. **a** Patient A's oxygen saturation ($SPO_2$) fluctuates significantly before dropping below 95% after around 150 h since her COVID-19 diagnosis, suggesting respiratory distress. The patient is subsequently intubated. This is followed by a sharp rise in serum creatinine levels, indicating potential acute kidney injury. Both SOFA and CovEWS reflect these events with an increase in Patient A's risk. Crucially, however, since CovEWS accounts for early deterioration in $SPO_2$ and white blood cell counts, it identifies the patient as high-risk much sooner than SOFA, triggering re-evaluation of current treatment strategy, including investigation for delayed complication or treatment injury, and/or the initiation of goals of care discussion. **b** In Patient B, different risk factors, including c-reactive protein (CRP), respiratory rate (RR) and $SPO_2$, weigh heavily in risk assessment. Initially, Patient B's RR increases significantly to over 30 breaths per minute while her $SPO_2$ drops below 95%, reflected by a corresponding increase in both SOFA and CovEWS. Patient B's RR and CRP levels however stabilise, which is correctly reflected in a lowering of the mortality risk by CovEWS. Intubation is averted for this patient. In contrast, SOFA does not account for the improvements in $SPO_2$, RR and does not reflect Patient B's improved state.

patients on June 5th 2020 and June 16th, respectively—which significantly impacted clinical treatment practice of COVID-19 patients. The COVID-19 diagnoses of all included patients were confirmed based on the presence of positive SARS-CoV-2 test results and/or COVID-19 diagnostic codes in their EHRs (Data Collection). The data formats were normalised across the two federated networks of HCOs (Data Collection), and all data were preprocessed to address the missingness that is characteristic for real-world clinical data (Preprocessing).

**Predictive performance for different prediction horizons.** We compared the predictive performance of CovEWS, several baselines and existing risk prediction scores ("Baselines"), including a version of CovEWS based on a linear time-varying Cox model (CovEWS [linear], "Time-varying Covariates"), COVID-19 Estimated Risk for Fatality (COVER_F)[19], Sequential Organ Failure

Assessment (SOFA)[14], Modified Early Warning Score (MEWS)[23], the decision tree developed by Yan et al.[17] and the deep learning model developed by Liang et al.[18], in terms of their respective specificity for identifying COVID-19 related mortality with a conservative fixed sensitivity of at least 95% and a slightly more relaxed level of 90% at a minimum of 1, 2, 4, 8, 16, 24, 48, 96 and 192 h (8 days) prior to observed mortality events on both the hold-out test data of Optum cohort and the external test cohort from the TriNetX network (Figs. 2). The last observed EHR entry's date was taken as a reference time for those patients that did not have an observed mortality event during the data collection period. In terms of specificity at a sensitivity greater than 95%, we found that CovEWS significantly ($p < 0.05$, one-sided Mann-Whitney-Wilcoxon with Bonferroni correction, see Supplementary Data 1 and Supplementary Data 2) outperformed other baselines and existing risk prediction scores at each

**Table 1 Descriptive statistics and percentage of patients with missing entries (Miss. %) for the training, validation and test sets of the Optum cohort and the external TriNetX test cohort.**

|  | Optum | | | | | | TriNetX | |
|  | Training set (March 21–June 5 2020) | | Validation set | | Test set | | External test set (March 21–June 25 2020) | |
|  | Value | Miss.% | Value | Miss.% | Value | Miss.% | Value | Miss.% |
|---|---|---|---|---|---|---|---|---|
| Patients [#] | 23,692 | – | 9477 | – | 14,215 | – | 5005 | – |
| COVID-19 [%] | 100.00 | – | 100.00 | – | 100.00 | – | 100.00 | 100.00 |
| Hispanic [%] | 11.90 | – | 11.44 | – | 12.18 | – | 5.57 | – |
| Black [%] | 23.33 | – | 23.94 | – | 23.30 | – | 37.74 | – |
| Caucasian [%] | 50.19 | – | 49.67 | – | 49.31 | – | 40.28 | – |
| Asian [%] | 3.41 | – | 3.43 | – | 3.61 | – | 3.30 | – |
| Inpatient admission [%] | 32.16 | – | 33.06 | – | 32.24 | – | n/a | 100.00 |
| ICU admission [%] | 7.12 | – | 7.13 | – | 7.13 | – | n/a | 100.00 |
| Mortality [%] | 5.34 | – | 5.38 | – | 5.38 | – | 6.91 | – |
| **Model inputs** Female [%] | 54.14 | 0.04 | 54.16 | 0.03 | 54.15 | 0.04 | 53.67 | 0.04 |
| Age [years] | 54.00 (27.00, 80.00) | – | 54.00 (27.00, 80.00) | – | 54.00 (27.00, 80.00) | – | 55.00 (30.00, 78.00) | – |
| Weight [kg] | 82.97 (58.97, 117.93) | 42.90 | 82.81 (58.23, 117.93) | 42.11 | 82.70 (58.94, 117.52) | 42.48 | n/a | 100.00 |
| Height [cm] | 167.64 (154.94, 182.88) | 47.27 | 167.64 (154.94, 182.88) | 46.74 | 167.64 (154.94, 182.88) | 46.68 | n/a | 100.00 |
| BMI [kg/m$^2$] | 29.32 (22.05, 40.64) | 46.13 | 29.32 (21.95, 40.71) | 45.65 | 29.26 (22.02, 40.74) | 45.64 | 28.19 (19.35, 36.57) | 79.02 |
| Intubation [%] | 4.23 | – | 4.22 | – | 4.24 | – | 9.53 | – |
| Temperature [°C] | 36.90 (36.45, 37.61) | 37.75 | 36.90 (36.43, 37.63) | 36.56 | 36.90 (36.46, 37.63) | 37.50 | 36.94 (36.28, 37.56) | 84.06 |
| SpO2 [%] | 96.33 (93.38, 99.00) | 35.50 | 96.33 (93.32, 99.00) | 34.45 | 96.29 (93.38, 99.00) | 35.26 | 95.00 (90.71, 98.00) | 82.44 |
| Heart rate [/min] | 84.33 (68.89, 102.67) | 38.14 | 84.36 (68.29, 102.17) | 37.04 | 84.70 (68.80, 102.93) | 37.46 | 84.56 (66.48, 103.00) | 79.82 |
| Respiratory rate [/min] | 18.59 (16.00, 24.00) | 42.74 | 18.57 (16.00, 24.00) | 41.87 | 18.67 (16.00, 24.00) | 42.21 | 18.50 (16.00, 24.93) | 75.42 |
| Dyspnoea [%] | 57.07 | – | 57.55 | – | 56.38 | – | 48.09 | – |
| Sys. blood pressure [mmHg] | 125.16 (108.22, 146.96) | 39.24 | 125.21 (108.69, 146.76) | 38.42 | 125.00 (108.31, 146.43) | 38.56 | 126.00 (107.00, 148.00) | 42.12 |
| Dias. blood pressure [mmHg] | 73.38 (61.67, 87.00) | 39.27 | 73.58 (61.69, 87.26) | 38.39 | 73.60 (61.69, 87.03) | 38.58 | 73.50 (60.10, 87.00) | 42.12 |
| Kidney disease [%] | 13.13 | – | 13.13 | – | 13.42 | – | 14.53 | – |
| Ischaemic heart disease [%] | 18.71 | – | 19.62 | – | 18.07 | – | 15.26 | – |
| Other heart diseases [%] | 53.55 | – | 54.01 | – | 54.06 | – | 56.36 | – |
| Cerebovascular disease [%] | 10.43 | – | 10.93 | – | 10.63 | – | 10.51 | – |
| Hypertension [%] | 46.10 | – | 45.85 | – | 46.16 | – | 48.37 | – |
| Diabetes [%] | 25.03 | – | 24.69 | – | 24.84 | – | 28.45 | – |
| Hyperlipidemia [%] | 40.25 | – | 40.53 | – | 40.56 | – | 38.26 | – |
| Cancer [%] | 21.48 | – | 20.92 | – | 21.41 | – | 21.80 | – |
| COPD [%] | 9.23 | – | 9.13 | – | 8.79 | – | 8.31 | – |
| Asthma [%] | 15.32 | – | 14.77 | – | 15.16 | – | 16.62 | – |
| Pulmonary embolism [%] | 2.98 | – | 3.01 | – | 2.69 | – | 3.66 | – |
| Connective tissue disease [%] | 2.56 | – | 2.73 | – | 2.72 | – | 4.38 | – |
| Inflamatory bowel disease [%] | 1.44 | – | 1.33 | – | 1.36 | – | 1.20 | – |
| Osteoarthritis [%] | 21.73 | – | 22.27 | – | 22.44 | – | 18.02 | – |
| Rheumatroid arthritis [%] | 27.59 | – | 29.01 | – | 27.81 | – | 21.14 | – |
| HIV [%] | 0.60 | – | 0.65 | – | 0.64 | – | 0.90 | – |
| Smoking (never) | 44.02 | 36.18 | 44.71 | 34.77 | 44.16 | 36.46 | 0.00 | 100.00 |
| Smoking (previous) | 15.48 | 36.18 | 16.44 | 34.77 | 15.15 | 36.46 | 0.00 | 100.00 |
| Smoking (current) | 4.32 | 36.18 | 4.08 | 34.77 | 4.22 | 36.46 | 0.00 | 100.00 |

**Table 1 (continued)**

| | Optum | | | | | | TriNetX | |
| --- | --- | --- | --- | --- | --- | --- | --- | --- |
| | March 21–June 5 2020 | | | | | | March 21–June 25 2020 | |
| | Training set | | Validation set | | Test set | | External test set | |
| | Value | Miss.% | Value | Miss.% | Value | Miss.% | Value | Miss.% |
| Smoking (unknown) | 0.00 | 36.18 | 0.00 | 34.77 | 0.00 | 36.46 | 0.00 | 100.00 |
| White blood cells [10*3/ul] | 6.94 (4.17, 12.34) | 49.24 | 6.96 (4.16, 12.26) | 48.57 | 6.98 (4.18, 12.45) | 48.40 | 7.12 (4.46, 13.08) | 96.72 |
| Neutrophil [%] | 70.50 (51.89, 84.25) | 50.67 | 71.00 (51.40, 84.65) | 50.08 | 70.98 (52.33, 84.43) | 49.87 | 72.10 (54.65, 85.48) | 97.16 |
| Lymphocytes [%] | 18.28 (7.50, 35.40) | 50.66 | 18.00 (7.20, 35.70) | 50.05 | 18.02 (7.50, 35.00) | 49.84 | 13.01 (5.28, 28.53) | 91.13 |
| Eosinophil [%] | 0.75 (0.00, 3.00) | 51.41 | 0.70 (0.00, 3.00) | 51.00 | 0.73 (0.00, 2.97) | 50.57 | 1.45 (0.15, 3.96) | 92.71 |
| Basophil [%] | 0.30 (0.00, 1.00) | 51.48 | 0.30 (0.00, 1.00) | 51.06 | 0.28 (0.00, 0.91) | 50.70 | 0.36 (0.10, 1.35) | 91.37 |
| Platelets [10*3/ul] | 232.38 (143.00, 365.00) | 49.28 | 233.21 (144.00, 368.10) | 48.62 | 232.68 (144.00, 366.63) | 48.45 | 220.09 (141.55, 345.90) | 96.76 |
| C-reactive protein [mg/l] | 73.62 (8.00, 185.52) | 68.39 | 71.50 (8.98, 184.92) | 67.98 | 73.38 (9.00, 183.01) | 67.81 | 69.29 (6.00, 200.12) | 55.18 |
| hs. C-reactive protein [mg/l] | 56.55 (4.85, 162.00) | 96.62 | 61.10 (3.43, 168.00) | 96.66 | 56.14 (4.96, 158.78) | 96.31 | 16.19 (3.38, 167.23) | 99.28 |
| Procalcitonin [ng/ml] | 0.16 (0.04, 2.59) | 82.33 | 0.16 (0.04, 2.82) | 81.96 | 0.17 (0.04, 2.59) | 81.99 | 0.24 (0.06, 3.08) | 79.98 |
| Fibrin D-dimer [mg/l] | 0.91 (0.29, 4.95) | 93.20 | 0.95 (0.31, 5.83) | 93.15 | 0.89 (0.26, 4.72) | 93.16 | 0.00 (0.00, 0.01) | 88.63 |
| Ferritin [ng/ml] | 574.00 (97.34, 2196.90) | 69.77 | 567.85 (101.02, 2211.85) | 69.59 | 566.49 (96.00, 2212.00) | 69.13 | 798.00 (155.52, 5625.30) | 62.10 |
| Cardiac Troponin T [ng/ml] | 0.02 (0.00, 0.15) | 69.57 | 0.02 (0.00, 0.18) | 68.94 | 0.02 (0.00, 0.16) | 69.05 | 0.01 (0.01, 0.09) | 88.57 |
| Creatinine [mg/dl] | 0.91 (0.62, 2.14) | 49.52 | 0.90 (0.61, 2.16) | 48.65 | 0.90 (0.60, 2.13) | 48.42 | 0.94 (0.62, 2.81) | 40.48 |
| Lactate dehydrogenase [U/l] | 331.33 (191.00, 606.87) | 70.94 | 335.00 (189.00, 605.71) | 70.94 | 330.94 (193.45, 590.83) | 70.55 | 348.00 (208.00, 703.67) | 61.22 |
| GGT [U/l] | 54.50 (13.25, 280.90) | 96.62 | 47.50 (13.30, 243.20) | 96.55 | 54.50 (14.57, 318.00) | 96.69 | 97.50 (20.80, 644.10) | 99.32 |
| AAT [U/l] | 33.67 (17.00, 85.00) | 54.60 | 34.00 (18.00, 88.39) | 53.87 | 33.67 (17.27, 87.40) | 53.77 | 39.15 (19.41, 96.38) | 47.33 |
| Creatine kinase [U/l] | 124.81 (38.00, 660.14) | 78.95 | 124.04 (37.00, 803.53) | 78.61 | 126.00 (37.00, 722.20) | 78.24 | 148.00 (37.49, 884.45) | 81.62 |
| Bilirubin [mg/dl] | 0.50 (0.30, 1.00) | 54.59 | 0.50 (0.30, 1.00) | 53.84 | 0.50 (0.30, 1.00) | 53.65 | 0.50 (0.30, 1.06) | 46.65 |
| Albumin [g/dl] | 3.42 (2.50, 4.40) | 54.41 | 3.42 (2.50, 4.40) | 53.70 | 3.44 (2.50, 4.40) | 53.48 | 3.27 (2.15, 4.20) | 62.60 |
| IL-6 [pg/ml] | 24.42 (6.57, 168.48) | 95.53 | 23.00 (7.00, 145.60) | 95.60 | 24.33 (7.00, 171.30) | 95.50 | 48.62 (8.86, 526.97) | 87.77 |
| pH | 7.40 (7.28, 7.47) | 84.29 | 7.40 (7.28, 7.47) | 85.07 | 7.40 (7.28, 7.47) | 84.29 | 7.41 (7.33, 7.47) | 86.77 |
| PCO2 [mmHg] | 40.91 (31.00, 56.21) | 88.03 | 41.00 (31.00, 55.93) | 88.34 | 41.06 (30.37, 56.00) | 88.05 | 40.36 (31.35, 51.01) | 90.85 |
| PaO2 [mmHg] | 88.26 (61.00, 131.00) | 88.08 | 87.18 (59.00, 127.26) | 88.35 | 88.00 (60.00, 130.91) | 88.10 | 86.23 (61.00, 135.97) | 91.35 |
| HCO3 [mmol/l] | 25.00 (19.50, 30.21) | 78.57 | 25.00 (19.60, 30.00) | 79.04 | 25.00 (19.88, 30.00) | 78.22 | 25.11 (20.03, 30.66) | 80.98 |
| CO2 [mmol/l] | 24.50 (20.50, 28.33) | 51.10 | 24.44 (20.50, 28.19) | 49.95 | 24.40 (20.56, 28.33) | 49.98 | 24.25 (20.51, 28.40) | 58.08 |

Input covariates of CovEWS are placed towards the bottom of the table and separated from covariates that are not inputs by a horizontal line. For binary covariates, the Value columns indicate the percentage of patients presenting with the condition at the end of their respective observation periods. For continuous measurements, the Value columns indicate the median and 10th and 90th percentiles in parentheses of the observed value for measurements that are collected once per patient, such as age, and the median of observed values for measurements that are collected multiple times per patient, such as heart rate. Supplementary Table 3 presents the ICD codes corresponding to the shown diagnoses. BMI body mass index, HIV human immunodeficiency virus, COPD chronic obstructive pulmonary disease, GGT gamma glutamyl transferase, AAT aspartate aminotransferase, IL6 Interleukin 6, n/a not available.

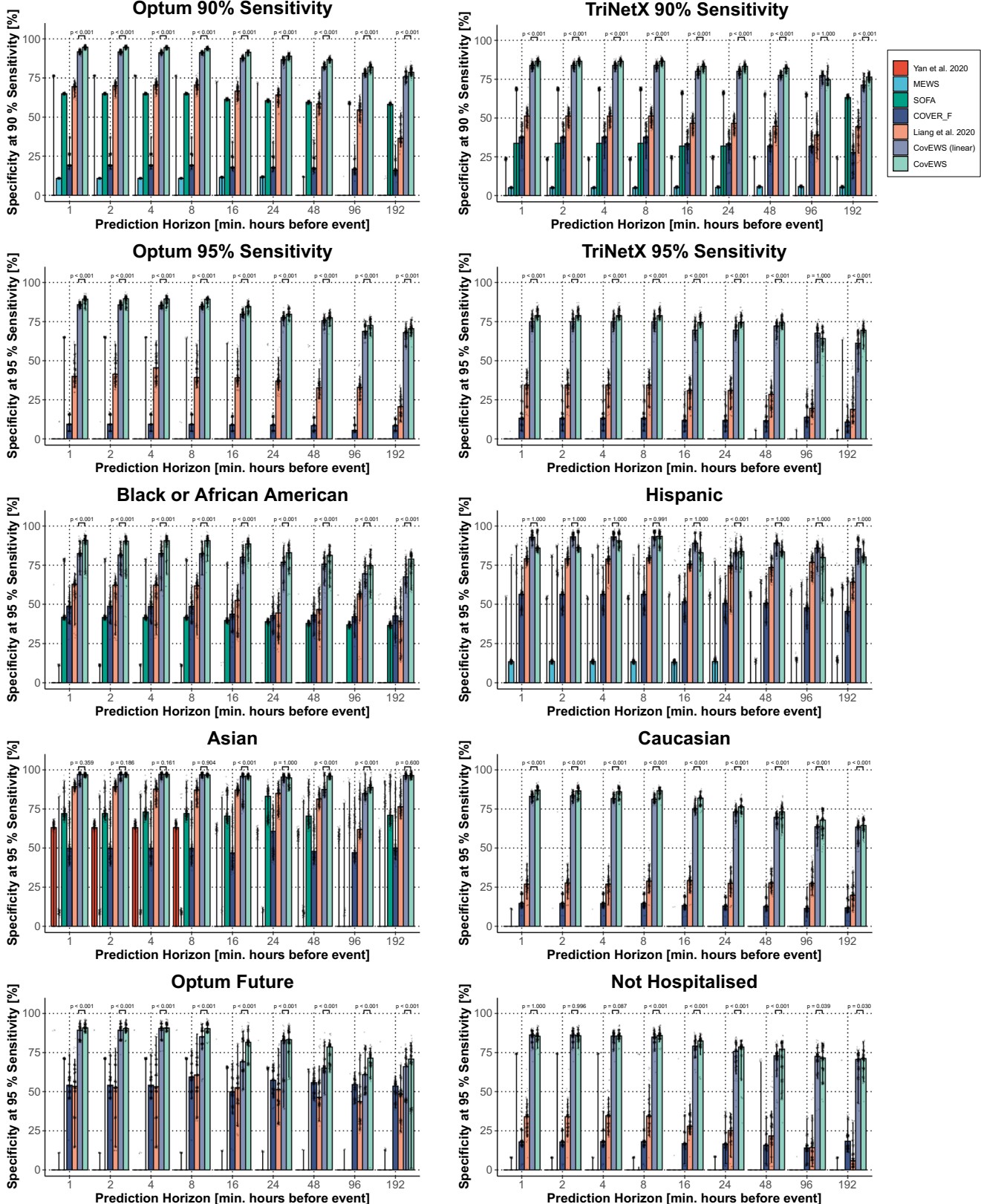

prediction horizon and on both the Optum and TriNetX cohorts with few exceptions. By comparing the predictive performances of the mortality prediction scores at different time horizons, we additionally quantified the degree to which risk prediction methods give more accurate predictions when the mortality event is closer to the prediction date. For example, the predictive

performance of CovEWS in terms of specificity at a sensitivity greater than 95% dropped from 89.3% (95% confidence interval [CI]: 83.0, 91.6%) to 70.5% (95% CI: 65.6, 76.4%) and from 78.8% (95% CI: 76.0, 84.7%) to 69.4% (95% CI: 57.6, 75.2%) from 1 h to 192 h prior to an observed mortality event on the held-out Optum test cohort and the external TriNetX test cohort,

**Fig. 2 Performance comparison in terms of Specificity at greater than either 90% (topmost row) or 95% (other rows) Sensitivity (*y*-axis) for different prediction horizons ahead of observed mortality events (in hours, *x*-axis) for CovEWS (light green), CovEWS (linear; light purple), Liang et al. (orange)[18], COVID-19 Estimated Risk for Fatality (COVER_F; blue)[19], Sequential Organ Failure Assessment (SOFA; green)[14], Modified Early Warning Score (MEWS; turquoise)[23], and Yan et al. (red)[17] on the held-out Optum test set, the external TriNetX test set, and selected patient subgroups from the Optum test set.** Some methods do not reach 90% and 95% sensitivity for some horizons, and may therefore not be visible in all plots. Bars indicate median and error bars indicate 95% confidence intervals (CIs) obtained via bootstrapping with 200 samples. Detailed results are available in "Performance Evaluation''. One-sided Mann-Whitney-Wilcoxon tests were used to derive p values shown at the top of each plot for superiority of CovEWS over CovEWS [linear].

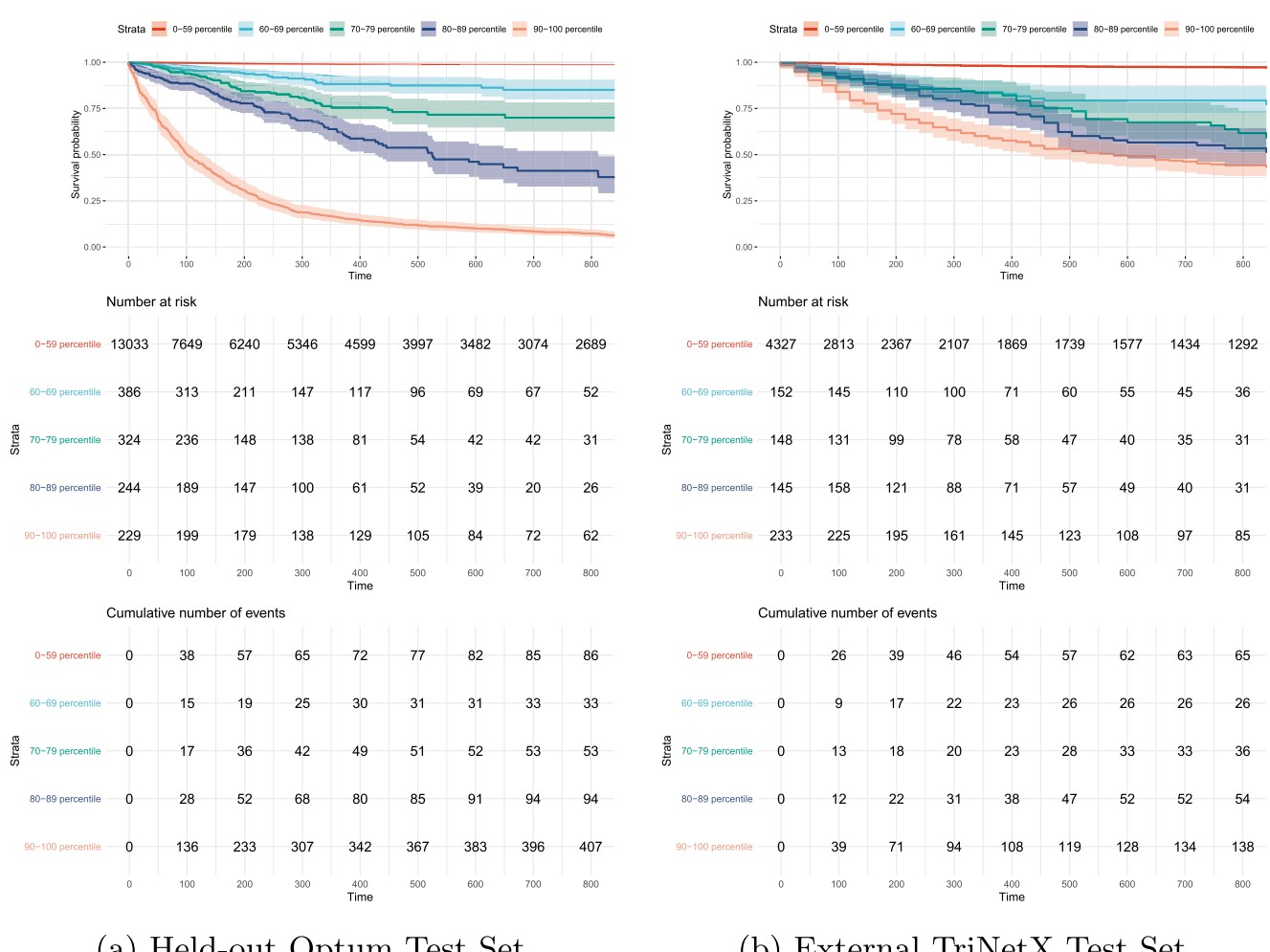

(a) Held-out Optum Test Set    (b) External TriNetX Test Set

**Fig. 3 Stratified survival analysis.** Stratification of patients in **a**. the held-out Optum test cohort (left, 14,215 patients) and **b**. the external TriNetX test cohort (right, 5005 patients) according to their assigned CovEWS score over time (in hours since COVID-19 diagnosis) into those patients that were assigned a CovEWS score below 60 (orange, bottommost), from 60 to 69 (deep blue), 70 to 79 (green), 80 to 89 (turquoise), and 90 to 100 (red, topmost). Shaded areas indicate 95% CIs calculated on the logarithmic scale from the standard errors of the Kaplan–Meier estimator with the centre values corresponding to the the Kaplan–Meier estimates[44]. Note that the five strata and their respective limits were chosen for clarity of visualisation—other strata are possible, and may, depending on context, have better clinical utility. Rows show time-varying survival probabilities (top row), the number of patients (centre row), and the cumulative number of mortality events observed (bottom row) for patients in each stratum of assigned CovEWS scores. Steeper curves indicate that more patients died while assigned a CovEWS score in the respective stratum. In contrast to traditional survival curves, cohorts as defined by strata of CovEWS scores are not static over time, and patients move between the stratified groups as they are assigned lower or higher CovEWS scores in response to their status improving or deteriorating, respectively. The results showed that CovEWS enables effective stratification of patients into risk groups over the course of their disease, as patients that were assigned a higher CovEWS score were more likely to die over time on both test cohorts while maintaining separation between the stratified cohorts.

respectively. When comparing the predictive performance across the held-out Optum test cohort and the external TriNetX test cohort, we saw the same trends in performance. However, all methods were roughly 10% less specific at greater than 95% sensitivity. This difference persisted even in those risk assessment systems that were not originally trained on the Optum training cohort, such as COVER_F. We thus attributed this apparent difference in performance not to overfitting to the Optum training cohort, but to (i) the difference of 5.38% against 6.91% in baseline mortality between the held-out Optum test cohort and the external TriNetX test cohort, respectively, and (ii) the higher degree of missingness in short-term mortality risk factors, such

as, e.g., respiratory rate, $SpO_2$ and blood pressure, in TriNetX (Table 1). In addition to assessing predictive performance, we also evaluated the calibration[24] of the risk scores predicted by Cov-EWS. We found that CovEWS overestimates mortality risk when interpreted as the probability of a mortality event occurring within the next 24 h because patients' states may change between the prediction time and the end of the prediction horizon (Preprocessing).

**Predictive performance for different subgroups**. We also compared the predictive performance of CovEWS against the baselines and existing scores across various ethnic subgroups, on patients that were not hospitalised, and on the Optum future cohort (Fig. 2; cohort statistics in Table 2). Overall, across each of these cohorts, we found that CovEWS significantly ($p < 0.05$, one-sided Mann-Whitney-Wilcoxon with Bonferroni correction) outperformed all of the baselines at each prediction horizon with the sole exception being the 96 and 192 h prediction horizons on the Optum future cohort—where the performance difference was not in all cases significant. The predictive performance difference between CovEWS and other risk assessment methods was more pronounced in the Caucasian and African American subgroups than in the Asian subgroup, which is likely reflective of the fact that several baselines have been developed using data from predominantly Asian populations. On the subgroup of patients that was not hospitalised, we found that, although lower than the overall performance on the entire Optum test set, CovEWS maintained a high level of performance. We attributed the lower performance on the non-hospitalised group compared to the overall Optum test set to (i) the considerably higher missingness in this patient group caused by non-hospitalised patients not being monitored as closely as hospitalised patients (Supplementary Table 4), and (ii) the overall considerably lower mortality rate in this patient group. Respectable performance on the non-hospitalised patient group is particularly important since the majority of COVID-19 patients are treated in an outpatient setting. In addition, when evaluating the various risk assessment methods on the Optum future cohort, we found that CovEWS was largely robust to changes in treatment policies and other temporal effects. A notable anomaly was the 96 and 192 h prediction horizons where the variance in our performance estimates was relatively high since fewer patients with recorded mortality outcomes and long-term monitoring data were available due to the shorter data collection time (5 weeks) of the Optum future cohort compared to the Optum test set (11 weeks) and the TriNetX test set (13 weeks). CovEWS likely remained robust to changes in treatment protocols because they improved outcomes by reducing the occurrence of critical patient states.

**Stratified time-varying survival analysis**. As illustrated in the examples in Fig. 1, CovEWS continuously varies over time since it accounts for the status of patients deteriorating or improving. To add to the analysis of the predictive performance of CovEWS in identifying the mortality of individual patients at fixed prediction horizons prior to observed mortality events presented in the previous paragraph, we therefore additionally evaluated whether CovEWS enables stratification of high-risk patients continuously over time (Fig. 3). To do so, we stratified the held-out Optum test cohort and the external TriNetX cohort into five strata of the CovEWS score, respectively, assigned to each patient ("Predictive Performance for Different Subgroups"). We found that CovEWS effectively separated patients into risk groups with distinct COVID-19 related mortality risk profiles, as patients that were assigned to higher strata of CovEWS scores were more likely to die across all strata over the course of their disease. When

comparing stratification results between the held-out Optum test cohort and the external TriNetX cohort, we observed that the ability to stratify patients into risk groups generalised across the two datasets—indicating that the predictive performance of CovEWS can transfer to other sources of data collected with different protocols, from different locations, and under different treatment policies. We also observed that the highest risk stratum of patients assigned CovEWS scores between 90 and 100 was considerably steeper than other strata in the held-out Optum test cohort and this anomaly did not persist to the same degree in the external TriNetX cohort. Qualitatively, we reasoned that this difference between the two datasets was due to the considerably higher missingness of short-term risk factors associated with mortality, such as, e.g., respiratory rate, $SpO_2$ and blood pressure, in the TriNetX cohort (Table 1). Rapid changes in these short-term risk factors often result in substantially increased near-term mortality risk and CovEWS scores reflected this increased risk immediately (Fig. 1), moving patients with extreme short-term risk indicators into the highest risk stratum. Since these short-term risk factors were not included as frequently in the TriNetX cohort, CovEWS was considerably less able to react to short-term deteriorations in the status of the patients, which was reflected in a relatively flatter time-varying survival curve of the highest risk stratum in the TriNetX cohort.

## Discussion
We developed and validated CovEWS, a real-time early warning system for predicting mortality of COVID-19 positive patients, using routinely collected clinical measurements and laboratory results from EHRs. Our validation aimed to evaluate CovEWS and other existing prediction systems for COVID-19 related mortality risk prediction in real-world conditions, which include among others missingness, variations in treatment policies and differences in treated populations at different sites, as observed in a large and representative cohort across multiple hospitals. When compared to existing prediction systems, our method not only provides accurate mortality predictions for each patient, but also provides a real-time early warning system of up to 192 h (8 days) prior to an observed mortality event for individuals, while identifying clinically-relevant factors for predictive performance. These results are sustained across various ethnic groups and cohorts. Notably, in comparison to existing mortality risk scoring systems, our method achieves significantly higher performance in terms of specificity at greater than 95% sensitivity across all evaluated prediction time frames, and generalises well to data collected under different treatment and data collection policies and environmental conditions. The implications of providing such an early warning system are significant. The provided risk assessment could potentially broadly aid in clinical decision-making, as well as in the prioritisation of care and resource allocation. More specifically, CovEWS could enable clinicians to intensify monitoring and therefore initiate treatments earlier in patients with a higher risk of mortality. Moreover, as an additional information source, CovEWS could also help clinicians to decide when to initiate palliative care to improve the quality of remaining life for patients with this need. Additional studies investigating if and how CovEWS can influence clinical decision-making would be necessary to improve both treatment outcomes, the involvement of palliative care, or resource allocation to reduce COVID-related mortality.

Before applying CovEWS in clinical practice, it is important to decide and calibrate appropriate warning thresholds, e.g., at the 85%, 90% or 95% sensitivity level (Thresholds). Especially when hospitals are overwhelmed and need to strictly allocate resources, alarm fatigue due to ill-calibrated thresholds ought to be

**Table 2 Descriptive statistics for selected subgroups (Caucasian, Asian, Black or African American, Hispanic, Not Hospitalised) of the held-out Optum test cohort, and for the Optum future cohort.**

| | Held-out optum test set | | | | | Future cohort |
| | Black | Hispanic | Asian | Caucasian | Not hospitalised | June 6–July 13 2020 |
| | Value | Value | Value | Value | Value | Value |
|---|---|---|---|---|---|---|
| Patients [#] | 3312 | 1732 | 513 | 7010 | 9366 | 14,041 |
| COVID-19 [%] | 100.00 | 100.00 | 100.00 | 100.00 | 100.00 | 100.00 |
| Hispanic [%] | 2.48 | 100.00 | 2.92 | 7.89 | 10.09 | 15.53 |
| Black [%] | 100.00 | 4.73 | 0.00 | 0.00 | 20.93 | 18.97 |
| Caucasian [%] | 0.00 | 31.93 | 0.00 | 100.00 | 51.70 | 49.47 |
| Asian [%] | 0.00 | 0.87 | 100.00 | 0.00 | 3.75 | 2.90 |
| Inpatient admission [%] | 36.78 | 44.69 | 31.19 | 29.60 | 0.00 | 24.44 |
| ICU admission [%] | 9.87 | 6.87 | 5.07 | 7.02 | 0.00 | 3.64 |
| Mortality [%] | 5.10 | 3.06 | 4.87 | 6.36 | 2.65 | 0.80 |
| Model inputs Female [%] | 60.27 | 51.79 | 54.00 | 53.88 | 56.54 | 54.83 |
| Age [years] | 52.00 (28.00, 76.00) | 45.00 (22.00, 70.00) | 48.00 (28.00, 74.00) | 58.00 (29.00, 84.00) | 52.00 (27.00, 79.00) | 44.00 (20.00, 73.00) |
| Weight [kg] | 89.45 (63.50, 125.35) | 79.78 (56.76, 108.86) | 68.04 (52.64, 93.31) | 82.78 (58.21, 117.52) | 82.37 (58.61, 117.03) | 81.40 (56.70, 115.18) |
| Height [cm] | 168.91 (157.48, 183.00) | 165.10 (152.40, 177.80) | 162.56 (152.40, 175.26) | 167.64 (154.94, 182.88) | 167.64 (154.94, 182.88) | 167.64 (153.67, 182.88) |
| BMI [kg/m$^2$] | 31.40 (23.00, 43.66) | 29.75 (22.88, 39.44) | 26.16 (20.74, 32.21) | 28.93 (21.81, 40.10) | 29.12 (21.95, 40.39) | 28.80 (21.52, 39.43) |
| Intubation [%] | 5.01 | 3.23 | 5.85 | 4.15 | a1.75 | 0.49 |
| Temperature [°C] | 36.94 (36.50, 37.65) | 36.90 (36.45, 37.70) | 36.90 (36.50, 37.80) | 36.85 (36.40, 37.62) | 36.90 (36.40, 37.62) | 36.83 (36.40, 37.68) |
| SpO$_2$ [%] | 96.81 (94.00, 99.00) | 97.00 (93.75, 99.00) | 96.50 (93.06, 99.00) | 96.00 (93.04, 98.59) | 96.82 (93.45, 99.00) | 97.52 (94.61, 99.14) |
| Heart rate [/min] | 85.86 (70.32, 103.33) | 86.14 (71.58, 106.17) | 86.22 (70.10, 103.48) | 83.00 (67.00, 101.00) | 84.00 (68.00, 103.00) | 81.82 (65.84, 101.00) |
| Respiratory rate [/min] | 18.68 (16.00, 24.00) | 18.57 (16.00, 24.00) | 19.00 (16.00, 24.99) | 18.63 (16.00, 23.70) | 18.00 (16.00, 22.88) | 17.79 (15.27, 21.00) |
| Dyspnoea [%] | 61.50 | 51.50 | 52.44 | 59.49 | 52.34 | 45.17 |
| Sys. blood pressure [mmHg] | 127.87 (111.57, 150.00) | 124.18 (107.00, 145.84) | 121.32 (103.58, 143.57) | 124.67 (108.34, 145.26) | 125.16 (108.00, 146.00) | 124.60 (107.86, 146.23) |
| Dias. blood pressure [mmHg] | 75.30 (63.36, 89.21) | 74.02 (62.00, 87.33) | 73.89 (61.02, 87.61) | 72.38 (61.00, 86.00) | 74.52 (62.00, 87.96) | 75.33 (63.00, 88.22) |
| Kidney disease [%] | 18.39 | 7.39 | 7.99 | 14.47 | 10.39 | 7.64 |
| Ischaemic heart disease [%] | 18.75 | 10.10 | 13.84 | 21.97 | 14.33 | 13.14 |
| Other heart diseases [%] | 62.98 | 39.38 | 43.86 | 59.39 | 47.92 | 41.97 |
| Cerebrovascular disease [%] | 11.35 | 5.31 | 7.99 | 13.27 | 8.58 | 7.63 |
| Hypertension [%] | 57.49 | 32.91 | 34.11 | 50.07 | 40.85 | 33.77 |
| Diabetes [%] | 33.24 | 23.27 | 22.81 | 23.79 | 20.71 | 18.07 |
| Hyperlipidemia [%] | 41.67 | 29.56 | 34.89 | 48.15 | 37.88 | 32.67 |
| Cancer [%] | 23.19 | 13.39 | 12.87 | 26.73 | 20.05 | 18.55 |
| COPD [%] | 9.09 | 4.16 | 2.73 | 11.57 | 6.74 | 5.70 |
| Asthma [%] | 19.72 | 14.67 | 10.33 | 15.82 | 13.40 | 13.27 |
| Pulmonary embolism [%] | 3.74 | 1.27 | 1.36 | 2.87 | 1.85 | 1.56 |
| Connective tissue disease [%] | 3.47 | 1.96 | 1.17 | 3.17 | 2.21 | 2.17 |
| Inflammatory bowel disease [%] | 1.00 | 0.40 | 0.58 | 2.00 | 1.33 | 1.18 |
| Osteoarthritis [%] | 25.39 | 12.18 | 10.92 | 27.92 | 20.51 | 17.41 |
| Rheumatoid arthritis [%] | 36.11 | 20.15 | 15.20 | 32.13 | 25.43 | 24.04 |
| HIV [%] | 1.03 | 0.81 | 0.19 | 0.46 | 0.49 | 0.48 |
| Smoking (never) | 46.23 | 56.12 | 53.22 | 40.30 | 38.11 | 53.28 |
| Smoking (previous) | 15.76 | 10.22 | 8.58 | 18.13 | 12.53 | 14.91 |
| Smoking (current) | 6.16 | 3.87 | 3.12 | 3.88 | 3.07 | 5.44 |
| Smoking (unknown) | 0.00 | 0.00 | 0.00 | 0.00 | 0.00 | 0.00 |

**Table 2 (continued)**

| | Held-out optum test set | | | | | Future cohort |
| | Black | Hispanic | Asian | Caucasian | Not hospitalised | June 6-July 13 2020 |
| | Value | Value | Value | Value | Value | Value |
|---|---|---|---|---|---|---|
| White blood cells [10*3/ul] | 6.75 (3.95, 12.33) | 6.93 (4.29, 12.25) | 7.06 (4.52, 12.35) | 7.00 (4.24, 12.28) | 6.80 (4.17, 11.90) | 6.60 (4.20, 11.00) |
| Neutrophil [%] | 71.00 (50.50, 84.74) | 70.36 (54.00, 83.62) | 73.31 (54.24, 85.30) | 71.00 (53.31, 84.05) | 69.00 (50.60, 84.33) | 63.50 (46.52, 80.50) |
| Lymphocytes [%] | 18.05 (7.44, 37.02) | 19.09 (8.17, 34.98) | 17.00 (6.98, 35.06) | 17.67 (7.41, 34.00) | 19.45 (7.53, 36.80) | 24.80 (10.50, 40.80) |
| Eosinophil [%] | 0.49 (0.00, 2.40) | 0.57 (0.00, 2.66) | 0.70 (0.00, 2.25) | 0.95 (0.00, 3.00) | 0.90 (0.00, 3.20) | 1.25 (0.00, 4.00) |
| Basophil [%] | 0.30 (0.00, 1.00) | 0.25 (0.00, 0.70) | 0.18 (0.00, 0.70) | 0.30 (0.00, 1.00) | 0.31 (0.00, 1.00) | 0.42 (0.00, 1.00) |
| Platelets [10*3/ul] | 236.00 (148.38, 369.84) | 236.00 (150.40, 368.80) | 242.33 (143.58, 348.74) | 229.00 (141.20, 352.53) | 234.33 (144.00, 362.94) | 235.42 (151.00, 343.00) |
| C-reactive protein [mg/l] | 70.00 (11.29, 172.50) | 69.62 (5.75, 188.17) | 79.08 (21.48, 176.79) | 73.83 (9.00, 183.42) | 69.12 (6.00, 179.19) | 40.10 (2.75, 158.30) |
| hs. C-reactive protein [mg/l] | 56.90 (9.20, 125.00) | 56.55 (3.07, 138.76) | 60.42 (3.95, 167.75) | 57.00 (4.50, 170.20) | 52.45 (3.26, 144.39) | 9.10 (0.80, 122.33) |
| Procalcitonin [ng/ml] | 0.18 (0.04, 3.84) | 0.15 (0.04, 2.84) | 0.32 (0.04, 2.83) | 0.16 (0.04, 1.90) | 0.19 (0.04, 2.96) | 0.13 (0.04, 1.38) |
| Fibrin D-dimer [mg/l] | 1.14 (0.32, 7.28) | 0.53 (0.18, 2.40) | 0.57 (0.25, 3.28) | 0.93 (0.29, 3.70) | 0.72 (0.24, 3.91) | 0.79 (0.31, 3.50) |
| Ferritin [ng/ml] | 574.83 (113.85, 2446.07) | 554.47 (88.00, 1956.78) | 911.01 (152.40, 3397.40) | 540.20 (90.23, 2015.77) | 533.73 (78.25, 2046.47) | 298.20 (33.38, 1577.72) |
| Cardiac Troponin T [ng/ml] | 0.01 (0.00, 0.13) | 0.01 (0.00, 0.09) | 0.03 (0.00, 0.49) | 0.02 (0.00, 0.19) | 0.03 (0.01, 0.20) | 0.01 (0.00, 0.09) |
| Creatinine [mg/dl] | 1.01 (0.67, 2.90) | 0.80 (0.53, 1.85) | 0.87 (0.57, 2.02) | 0.89 (0.61, 1.88) | 0.90 (0.61, 1.88) | 0.86 (0.60, 1.44) |
| Lactate dehydrogenase [U/l] | 347.00 (209.28, 602.25) | 325.18 (183.77, 616.27) | 394.50 (217.40, 719.21) | 312.00 (186.50, 558.85) | 328.00 (189.96, 586.03) | 282.71 (171.00, 567.93) |
| GGT [U/l] | 40.67 (20.00, 287.43) | 69.90 (25.00, 538.32) | 73.02 (25.05, 392.52) | 43.00 (11.00, 247.54) | 58.00 (13.00, 315.43) | 30.00 (10.25, 202.75) |
| AAT [U/l] | 34.00 (18.00, 85.45) | 36.42 (18.00, 88.82) | 44.67 (22.07, 133.48) | 31.33 (17.00, 81.94) | 30.54 (17.00, 80.41) | 24.00 (15.00, 56.30) |
| Creatine kinase [U/l] | 171.35 (49.50, 908.83) | 105.29 (40.00, 497.77) | 133.20 (42.94, 643.30) | 100.47 (31.23, 593.24) | 127.00 (41.00, 641.50) | 110.00 (36.00, 564.10) |
| Bilirubin [mg/dl] | 0.50 (0.28, 1.00) | 0.50 (0.30, 0.90) | 0.58 (0.30, 1.05) | 0.50 (0.30, 1.00) | 0.50 (0.30, 0.90) | 0.45 (0.20, 0.90) |
| Albumin [g/dl] | 3.43 (2.56, 4.20) | 3.50 (2.55, 4.40) | 3.40 (2.30, 4.40) | 3.43 (2.50, 4.50) | 3.62 (2.70, 4.60) | 4.10 (2.90, 4.70) |
| IL-6 [pg/ml] | 19.00 (6.48, 178.40) | 22.00 (6.00, 233.17) | 31.00 (7.88, 206.45) | 21.67 (7.00, 165.00) | 28.00 (7.00, 207.53) | 8.10 (4.20, 52.49) |
| pH | 7.40 (7.29, 7.48) | 7.40 (7.27, 7.47) | 7.41 (7.25, 7.46) | 7.40 (7.28, 7.48) | 7.39 (7.26, 7.47) | 7.42 (7.31, 7.48) |
| $PCO_2$ [mmHg] | 40.38 (29.63, 53.60) | 41.73 (32.00, 56.99) | 42.04 (32.32, 57.66) | 41.40 (30.00, 55.91) | 40.95 (30.00, 57.89) | 37.00 (28.25, 50.64) |
| $PaO_2$ [mmHg] | 87.30 (61.06, 136.69) | 87.47 (60.18, 126.00) | 97.29 (66.18, 129.43) | 86.64 (59.00, 130.00) | 86.78 (60.07, 141.46) | 77.00 (53.00, 130.50) |
| $HCO_3$ [mmol/l] | 25.00 (19.97, 29.70) | 25.00 (20.00, 30.25) | 24.62 (18.52, 29.20) | 25.00 (20.00, 30.53) | 25.00 (20.00, 29.48) | 24.67 (19.30, 29.00) |
| $CO_2$ [mmol/l] | 24.20 (20.33, 28.26) | 24.42 (20.87, 28.07) | 24.00 (19.39, 27.04) | 24.59 (20.86, 28.71) | 24.50 (20.75, 28.50) | 24.13 (21.00, 28.00) |

Input covariates of CovEWS are placed towards the bottom of the table and separated from covariates that are not inputs by a horizontal line. For binary covariates, the Value columns indicate the percentage of patients presenting with the condition at the end of of their respective observation periods. For continuous measurements, the Value columns indicate the median and 10th and 90th percentiles in parentheses of the observed value for measurements that are collected once per patient, such as age, and the median of observed values for measurements that are collected multiple times per patient, such as heart rate.
BMI body mass index, HIV human immunodeficiency virus, COPD chronic obstructive pulmonary disease, GGT gamma glutamyl transferase, AAT aspartate aminotransferase, IL6 interleukin 6, n/a not available.
[a]see "Data Collection" for an explanation of the non-zero intubation rate.

minimised. Beyond thresholds, it is also important to choose appropriate integration points with existing clinical workflows. While such integrations are highly dependent on existing guidelines and context, key timepoints along the patient journey, such as pre-testing, admission, discharge, prior to and after significant interventions such as intubations, and when monitored in critical care on a continuous basis, could be potentially sensible times to assess a patient's CovEWS scores. In addition, while the data used in this study already comprises multiple hospitals, a further analysis including hospitals from other countries would be useful to investigate the impact of geographic and cultural differences—particularly in those geographic contexts that are not well covered by this study. Due to differences in data collection methodology and expected data formats, another limitation of this study is that the implementation of some existing risk scoring systems is based on certain assumptions that may adversely influence their comparative performance (Baselines). Another limitation of this study related to data collection is that the death records in our datasets may not have been complete for all patients, because HCOs may not in all cases have been aware of deaths that happened to patients that were not under their care anymore. Additionally, the do not resuscitate (DNR) status of patients, which may have significant ramifications for their mortality[25], was not available as an input to CovEWS. Similarly, not all important covariates were available for all patients at all time points since our evaluation was based on patient data collected in real care environments, where missingness is pervasive (Table 1). To handle missingness, we employed multiple imputation by chained equations (MICE)[26] (Preprocessing) which was recommended by Gerry and colleagues[27] for handling missing information in early warning scoring systems. Since imputation errors may have influenced the performance of prediction models, we performed additional analyses on several subsets of patients that had fewer missing covariates to ensure the prediction performance of any specific prediction system was not disproportionately affected by imputation errors (Supplementary Note 1). While there are, in general, various important considerations in evaluating risk prediction systems that rely on different input covariates under missingness, we consider the presented evaluation approach to be the most representative for the envisioned potential clinical use of CovEWS (Supplementary Note 2). Beyond considerations around missingness, we note that this work only concerns risk scores from routinely collected clinical data and patients who are already seeking care at healthcare providers. For efficient mitigation of COVID-19, additional, potentially preventative efforts like tracking apps, risk scores of infection prior to admission, masks and social distancing are necessary.

It is also important to acknowledge upfront the pitfalls of mortality prediction of hospitalised patients. A significant proportion of patients who die in the hospital, do so after cessation of treatment. One may argue that models that predict mortality thus actually predict the likelihood of treatment discontinuation. Numerous factors go into the decision with regard to continuing or stopping interventions, including whether the outcome, if the patient were to survive, is aligned with the patient's preferences. It will only be accurate in a clinical context where clinicians make predictions in a similar manner, where patients share the same values and preferences around the quality of life, and where the decision-making process resembles that of the training cohort.

From the perspective of medical staff, prognostication, as well as the perception of the quality of life if the patient were to survive, determine the framing of patient status to the family and friends; these are vulnerable to bias, both conscious or unconscious and influence the decision to admit the patient to the intensive care unit, as well as the decision to discontinue treatment (which almost certainly lead to death among those who are most severely ill). In a perfect world without bias and health disparities, only patient and disease factors determine hospital mortality, but studies have repeatedly demonstrated that this is far from the case. Recently, mortality from critical illness has been shown to be higher in disproportionately minority-serving hospitals after adjustment for illness severity and other biological factors that pertain to the patient and to the disease[28,29]. It is nearly impossible to incorporate these factors precisely in a model that is trained on mortality as an outcome. As a decision support tool to inform discussion around goals of care, CovEWS is subject to the same limitations that mortality prediction models have—it may permeate or even magnify existing health disparities and provider bias. As an early warning system, however, we speculate that the impact of the exclusion of social determinants on model performance is acceptable.

In summary, we presented, developed, and experimentally validated CovEWS, a real-time early warning system that provides clinically meaningful predictions of COVID-19 related mortality up to 192 h (8 days) in advance for individual patients using routinely collected EHR data. In contrast to existing risk scoring systems, CovEWS provides real-time continuous risk assessment that accounts for a large set of short-term and long-term risk factors associated with COVID-19 related mortality, is automatically derived from readily available EHR data, and was externally validated using data from multiple hospitals, diverse patient groups, and across time frames. In terms of influential covariates, a number of known risk factors, including age, $SpO_2$, blood pressure, ischaemic and other heart diseases, hypertension, white blood cell count, neutrophils, lymphocytes, high-sensitivity c-reactive protein, creatinine, lactate dehydrogenase, albumin, pH and $PCO_2$, were found to be significantly associated with mortality outcomes in the Optum training fold (Table 2). Accessible risk assessment from readily available EHRs is especially important in the ongoing COVID-19 pandemic since access to advanced clinical lab testing and imaging techniques may be limited in many hospitals. CovEWS allows for critical time in clinical decision making, even without access to specialised lab tests or advanced diagnostic equipment. Prospective studies are needed to conclusively establish if the availability of early warnings for COVID-19 related mortality through CovEWS improves patient outcomes compared to the standard of care.

## Methods

**Overview**. The overall pipeline of the method is shown in Supplementary Fig. 1. We refer to "Nonlinear Time-varying Covariates" for a detailed presentation of the predictive model used by CovEWS and Supplementary Fig. 3 for a detailed diagram of the model architecture.

**Data collection**. We used data collected by two federated networks of healthcare organisations:

*Optum*. The Optum de-identified COVID-19 electronic health records database includes de-identified electronic medical records and clinical administrative data including bedside observations and laboratory data from a geographically diverse set of healthcare institutions in the United States (US). The EHR data was sourced from more than 45 provider groups and integrated delivery networks. We used Optum cohort data collected between 21st March and 5th June 2020, and another cohort separated in time from 6th June to 13th July 2020 for our analysis.

*TriNetX*. TriNetX is a global health research network providing a de-identified dataset of electronic medical records (diagnoses, procedures, medications, laboratory values, genomic information) including patients diagnosed with COVID-19. The data is de-identified based on standard defined in Section §164.514(a) of the Health Insurance Portability and Accountability Act (HIPAA) Privacy Rule. The process by which Data Sets are de-identified is attested to through a formal determination by a qualified expert as defined in Section §164.514(b)(1) of the HIPAA Privacy Rule. We used TriNetX cohort data collected between 21st March and 25th June 2020 from 24 healthcare organisations in the US, Australia, Malaysia and India for our analysis.

*Data quality*. Both Optum and TriNetX, as well as the data providing healthcare institutions applied quality control steps to their data, but these procedures are not standardised neither across the federated networks nor across healthcare institutions. Varying levels of data quality across EHRs collected at different healthcare institutions and networks are therefore expected. However, heterogeneous data quality standards are characteristic for real-world data collected at different healthcare institutions. By evaluating CovEWS against an external test cohort from healthcare institutions with data collection policies different from our training cohort, we are able to give a fair assessment as to how robust and transferable CovEWS is in presence of realistic variations in data quality.

*Inclusion criteria*. We only included patients that were COVID-19 positive in our analysis. In both datasets, we considered patients COVID-19 positive if they either (i) were diagnosed with any of the International Statistical Classification of Diseases and Related Health Problems 10th revision (ICD-10) codes J12.89, J20.8, J40, J22, J98.8 and J80 together with B97.29 or (ii) had a positive COVID-19 lab test result (Supplementary Table 1). The criteria listed in (i) correspond to the Centers for Disease Control and Prevention (CDC) COVID-19 coding guidelines effective February 20, 2020[30]. A sensitivity analysis performed in a recent epidemiological study using the Optum Research Database concluded that there were no significant differences in observed outcomes between COVID-19 patients that were included based on their recorded diagnoses and those included based on positive SARS-CoV-2 test results[31]. For patients identified as COVID-19 positive via ICD diagnosis codes, we used the date of diagnosis as the reference diagnosis date for our analyses. For those patients identified as COVID-19 positive via a positive lab test, we used the date of the test sample collection as the diagnosis date. For patients with both a positive COVID-19 lab test and diagnosis, the available diagnosis date took precedence. For the subgroup of patients that were not hospitalised, we included all patients that were neither admitted to a hospital as inpatients nor an intensive care unit (ICU) at any point according to their EHRs. We note that it is possible that hospitals did not in all cases record inpatient hospital admissions and ICU admissions in their respective EHRs—which may explain the observed non-zero rate of intubations in the non-hospitalised group. Membership in the Asian, Caucasian and Black or African American subgroups was mutually exclusive in the underlying EHR data model, and a patient could therefore only be assigned to one of the subgroups. In contrast, hispanic ethnicity was assigned in conjunction with any of the previous ethnic categorisations. We note that several ethnic subgroups were relatively small and therefore likely not representative of all patients in those groups.

*Feature selection*. We selected EHR-derived covariates for inclusion as input variables for CovEWS based on (i) previously published research on clinical risk factors for COVID-19[32–34], and (ii) expert input from several medical professionals involved in the treatment of COVID-19 patients. In addition, we aimed to include both short-term and long-term risk factors for COVID-19 related mortality due to the continuous real-time evaluation of CovEWS. We note that the do not resuscitate (DNR) status of patients, which may have significant ramifications for their mortality[25], was not included in our datasets. In our experimental evaluation, information from diagnostic codes was only used from the time point it was entered into the patient EHR. 85.0% and 72.5% of all diagnoses observed in the TriNetX and Optum cohorts, respectively, were available before inclusion of a patient into the cohort based on a COVID-19 diagnosis or positive SARS-CoV-2 test result. The remainder of diagnoses was added after the COVID-19 diagnosis of the patient. We present the list of all included model input covariates including their p-values in Supplementary Table 2, and their distributions across the datasets in Supplementary Table 1.

*Data characteristics*. The cohort statistics of the two datasets are presented in Supplementary Table 1, the ICD-9 and ICD-10 codes corresponding to the diagnoses shown in Supplementary Table 1 are given in Supplementary Table 3, and the number of observations per patient for time-varying covariates for the two datasets is visualised in Supplementary Fig. 2. As is characteristic for clinical data collected in real-world contexts, missing covariates are common in both datasets. Missingness in real-world EHR data is caused primarily by differences in laboratory testing guidelines, data collection practices, available testing resources and measurement devices between hospitals, and may in some cases depend on patient status and preferences of clinical staff. For example, Supplementary Table 4 compares the missingness between the Optum test set and the non-hospitalised patient subgroup of the Optum test set. In contrast to traditional clinical studies, realistic missingness patterns in both the training and evaluation datasets are a desirable feature in the context of our study as CovEWS is designed to be deployed in clinical contexts with similar missingness, and therefore has to be trained and evaluated in the presence of missingness patterns seen across a representative range of heterogeneous hospitals. Covariates were mostly balanced across the Optum and TriNetX datasets. The primary differences were a higher observed mortality rate, and higher ratios of intubations, connective tissue disease, and rheumatoid arthritis in the TriNetX data compared to the Optum data. In addition, we note that the majority of admissions were recorded as being of unknown type in the TriNetX database. Since the large fraction of unknown admission entries limited potential admission outcome analyses, we reported hospital and ICU admission outcomes as

not available for TriNetX (Supplementary Table 1). In compliance with the HIIPA Privacy Rule Section §164.514(a), patients' exact dates of death were not available to protect patient privacy. In our analysis, we therefore imputed the last recorded EHR entry date as the reference date of death for deceased patients. The actual dates of death may have happened at a later point, and our performance estimates are therefore potentially underestimating actual predictive performance, since (i) correct predictions that happened later would mean CovEWS predicted sooner than we thought for that patient (which is generally harder, see Supplementary Fig. 2), and (ii) incorrect predictions of CovEWS may actually have been outside of the prediction time horizon. We believe this approximation of the exact date of death is therefore an acceptable trade-off, since underestimation of performance is not as much a concern as overestimation would be.

**Data normalisation**. The EHR data across both data sources used two different, but compatible, underlying data models consisting of recorded diagnoses, demographics, lab tests, procedures, medications and clinical observations. For our risk factors of interest, we converted records from both datasets into a unified data representation. We used ICD-9 and ICD-10 to extract diagnoses (Supplementary Table 3), Logical Observation Identifiers Names and Codes (LOINC) to extract lab tests, Current Procedural Terminology (CPT), and ICD-9 Clinical Modification (ICD-9-CM) and ICD-10 Procedural Coding System (ICD-10-PCS) to extract intubation events from the EHR records. For lab tests, we additionally normalised the unit of each category of lab tests to be the same for each measured record of that category.

**Stratification**. We split the Optum cohort used for model development into training (50%), validation (20%) and held-out test folds (30% of all patients) at random stratified by patient age, gender, presence of mortality events, presence of intubation events, presence of ICU admission and presence of a human immunodeficiency virus (HIV) diagnosis. We added HIV to the set of stratification covariates since its low prevalence could otherwise have led to imbalances in this risk factor across the folds. Stratification produced balanced cohorts across the three folds (Supplementary Table 1). The Optum training fold was used to train CovEWS, the validation fold was used to select the optimal hyperparameter configuration for CovEWS, and the held-out test fold was used in addition to the external TriNetX test cohort to evaluate the out-of-sample generalisation performance of CovEWS.

**Preprocessing**. Discrete covariates with $p$ different values were transformed into their one-hot encoded representation with one out of $p$ indicator variables set to 1 to indicate the discrete value for this patient. All continuous features were standardised to have zero mean and unit standard deviation using observed covariate distribution on the Optum training fold. Missing values of continuous covariates were imputed in an iterative fashion using multiple imputation by chained equations (MICE)[26]. MICE was recently independently recommended by[27] for handling missing information in early warning scoring systems. MICE models were derived from the Optum training set, and at least 800 patients ($\approx$3.38% of the training set) were available for this purpose for all included covariates. After the preprocessing stage, continuous input features were standardised and fully imputed, and discrete input covariates were one-hot encoded. All preprocessing operations were derived only from the training fold, and naïvely applied without adjustment to other folds and datasets in order to avoid information leakage.

**Model**. We adopt a variation of the widely used Cox proportional hazard model that is adapted to accommodate nonlinear and time-varying effects of covariates on the log-hazard function. In the following, the basics of time-to-event analysis that is the main subject of this paper is briefly presented. Then we touch upon the Cox proportional model for continuous-time covariates that is followed by the modifications we applied to this model to prepare it for this work.

*Survival analysis*. Survival analysis which is also known as Time-To-Event (TTE) analysis included a large body of work consisting of mathematical tools to give a statistical analysis of the time duration until a specified event occurs. In this work, the event is defined to be the time when a patient dies.

An important tool in time-to-event analysis is *hazard function*. In discrete-time setting, (e.g., if times are given in specified periods) the hazard function is a conditional probability defined in discrete-time as

$$h(t|\mathbf{x}) = P(T = t | T \geq t; \mathbf{x}), \ t = 1, 2, \ldots \tag{1}$$

that represents the risk of dying at time $t$ if the patient has survived until that time. The relevant covariates of the patients up to time $t$ are encapsulated in the vector $\mathbf{x} \in \mathbb{R}^d$. Age, sex and lab tests are examples of such covariates that can take either binary or standardised real values after preprocessing. Intuitively, the hazard function captures the underlying dynamics of the transition of the condition of the patient from alive to dead. The larger $h$ is at time $t$, the more likely it is for the patient to die at time $t$.

Another useful function is called *survival function* that is denoted by $S(t)$ and in discrete-time defined as

$$S(t) = P(T > t) = \prod_{s=1}^{t}(1 - h_s). \tag{2}$$

Similar functions can be defined in the continuous-time regime. Let $T_c$ be the continuous survival time with the probability density function $f_c$ and cumulative distribution function $F_c$. Similar to Eq. (2), the continuous survival function represents the probability of surviving until time $t$ that is defined as

$$S_c(t|\mathbf{x}) = P(T_c > t|\mathbf{x}) = 1 - F_c(t|\mathbf{x}). \tag{3}$$

Likewise, the continuous hazard function is defined as

$$h_c(t|\mathbf{x}) = \lim_{\Delta t \to 0} \frac{P(t \le T_c \le t + \Delta t|T_c \ge t; \mathbf{x})}{\Delta t}. \tag{4}$$

Notice that unlike discrete hazard function Eq. (1), the continuous hazard function Eq. (4) is not a probability distribution and can take values larger than one.

The last useful function in continuous survival analysis is the cumulative hazard function

$$H_c(t) = \int_0^t h_c(u)du. \tag{5}$$

The connection between these quantities can be simply derived:

$$h_c(t) = \frac{f_c(t)}{S_c(t)}, \tag{6}$$

$$S_c(t) = \exp\left(-\int_0^t h_c(u)du\right) = \exp(-H_c(t)), \tag{7}$$

$$f_c(t) = h_c(t)\exp\left(-\int_0^t h_c(u)du\right) = h_c(t)\exp(-H_c(t)). \tag{8}$$

Before introducing the simple yet flexible Cox model, we discuss a few important issues that must be taken into account in survival analysis.

**Censoring**. What makes the survival analysis different from a simple regression from the covariates $\mathbf{x}$ to $T$—observed duration up to the occurrence of the event—is the concept of *censoring*. An observation is called censored—or more precisely *right-censored*—if its survival time has not been fully observed. There are several causes for a censored observation. For example, if a patient is not under observation when the event occurs or if the information of the patient is lost for some reason, only a lower bound to the time-to-event $T$ is observed that is the last time the condition of the patient is recorded.

**Discrete vs. continuous**. Although time is a continuous physical quantity, in practice, it is measured at discrete points. Especially, in medical sciences, the condition of the patient is measured on a regular daily or bi-daily basis. This implies that even though the change of the covariates of a patient occurs at certain points of time, the exact time is not known. The transition point is only known up to the resolution of the measurement. We assume the time at which an event of interest occurs is denoted by $T$. As the resolution of the measurement is hours in the datasets used in this work, $T = t$ refers to an event that occurs within the $t$th hour after the patient is admitted to the hospital and its health condition is recorded.

**Ties**. In the limited resolution measurement of time, some observations may have the same survival times, e.g., two patients die on the same day even though it is extremely unlikely that both die at the same moment. However, even in continuous time data, ties may occur which is a hint of underlying discrete sampling in time.

A major difference between continuous and discrete-time survival analysis is that the hazard function is a probability distribution in discrete settings while it can take any positive value in continuous settings. However, the traditional continuous-time approach can still be used for discrete event times especially when the measurements are equally spaced.

*Cox hazards model*. The most widely known model in the analysis of continuous survival time is Cox's proportional hazard model[35] that parameterises the hazard function as

$$h_c(t|\mathbf{x}) = h_0(t)\exp(\boldsymbol{\beta}^{\mathsf{T}}\mathbf{x}), \tag{9}$$

where $h_0$ is called the baseline hazard that is modulated by the effect of covariates via $\exp(\boldsymbol{\beta}^{\mathsf{T}}\mathbf{x})$. Notice that in the traditional Cox model Eq. (9) the covariates $\mathbf{x}$ are assumed constant over time. Consequently, the temporal variation of the hazard function is separated from the influence of the covariates.

*Time-varying covariates*. In many experimental settings, the assumption of time-invariant covariates in Eq. (9) does not hold. For example, many entries in the electronic health records such as heart rate, temperature, and blood measurements do not remain constant over the course of the hospitalisation of a patient. Therefore, the traditional Cox model Eq. (9) is extended to a time-varying setting by replacing $\mathbf{x}$ in Eq. (9) with $\mathbf{x}_t$ that is the measured covariates at time $t$. Assume a

dataset consists of $N$ patients indexed by $n = 1, 2, ..., N$. As a notational convention, $\mathbf{x}_t^{(n)}$ denotes the vector of the corresponding covariates to the patient $n$ at time $t$.

If the Cox model holds and continuous events are observed, the following function called *partial likelihood* is maximised to estimate $\boldsymbol{\beta}$:

$$\mathcal{L}(\boldsymbol{\beta}) := \prod_{i=1}^{k} \frac{\exp(\boldsymbol{\beta}^{\mathsf{T}}\mathbf{x}_{t_i}^{(i)})}{\sum_{j\in\mathcal{R}(t_i)}\exp(\boldsymbol{\beta}^{\mathsf{T}}\mathbf{x}_{t=t_i}^{(j)})}, \tag{10}$$

where $t_1 < t_2 < ... < t_k$ are the ordered times at which the events occur and $\mathbf{x}_{t_1}^{(1)}, \mathbf{x}_{t_2}^{(2)}, ..., \mathbf{x}_{t_k}^{(k)}$ are the corresponding set of covariates at those times. Notice that the equality of the superscript of the covariate vector $\mathbf{x}_{t_i}^{(i)}$ (patient's index) and the subscript of time $t_i$ emphasises the continuous event times and the fact that at most one patient experiences the event at each time. For the moment, we assume time is continuous that results in distinct event times. The set $\mathcal{R}(t_i)$ is the set of the patient's indices that are at risk at time $t_i$. Being at risk means they are alive and can potentially experience the event.

*Nonlinear time-varying covariates*. One clear limitation of Eq. (10) that is caused by the definition of the hazard function Eq. (9), is the fact that the exponent of the modulating function $\exp(\boldsymbol{\beta}^{\mathsf{T}}\mathbf{x})$ is a linear function of $\mathbf{x}$. Hence higher order interactions among different dimensions of the covariate vector cannot be captured by this method. To improve the expressiveness of the model, we replace the linear function $\boldsymbol{\beta}^{\mathsf{T}}\mathbf{x}$ with a nonlinear function realised by a neural network. Let $\phi(\cdot; \boldsymbol{\theta}) : \mathbb{R}^d \to \mathbb{R}$ be the function implemented by the neural network and parameterised by $\boldsymbol{\theta}$. Therefore, the hazard model is represented as

$$h(t|\mathbf{x}) = h_0(t)\exp(\phi(\mathbf{x}_t; \boldsymbol{\theta})), \tag{11}$$

where $h_0(t)$ is the baseline population-level hazard that is independent of the associated covariates to each patient. Time-varying covariates are transformed by the function $\phi(\cdot; \boldsymbol{\theta})$ to log-hazard. The parameters $\boldsymbol{\theta}$ are learned via maximising the partial log-likelihood[35]. Despite traditional Cox proportional hazard model where the gradient and Hessian can be computed analytically, here, we use automatic differentiation to compute gradients with respect to $\boldsymbol{\theta}$. The nonlinear function $\phi(\cdot; \boldsymbol{\theta})$ is implemented as a 2-layer multilayer perceptron—see "Model" for a detailed description. The hazard function Eq. (11) estimates the instantaneous risk of death at each time for each patient. Integrating with respect to time and exponentiating the result gives the survival function defined as

$$S(t|\mathbf{x}_{0:t}) = P(T > t|\mathbf{x}_{0:t}) = \exp\left(-\int_0^t h(u|\mathbf{x}_u)du\right). \tag{12}$$

Notice that $\mathbf{x}_{0:t}$ denotes the set of covariates until time $t$, meaning that, the probability of survival up to time $t$ depends on the history of the covariates.

The partial likelihood Eq. (10) is re-written as

$$\mathcal{L}(\boldsymbol{\theta}) := \prod_{i=1}^{k} \frac{\exp(\phi(\mathbf{x}_{t_i}^{(i)}; \boldsymbol{\theta}))}{\sum_{j\in\mathcal{R}(t_i)}\exp(\phi(\mathbf{x}_{t=t_i}^{(j)}; \boldsymbol{\theta}))}. \tag{13}$$

To give an intuition of Eq. (13), observe that the partial log-likelihood that is computed by taking logarithm of the right-hand side of Eq. (13) will consist of $k$ terms corresponding to $k$ observed events. The parameter vector $\boldsymbol{\theta}$ is perturbed such that the hazard increases for the covariates of a patient who dies at time $t_i$ while it decreases for the covariates of the patients who remain alive at $t_i$.

*Resolving ties*. Even though we adopt a continuous-time approach due to the non-normalised parametric form of the hazard function Eq. (9) and the resultant partial likelihood Eq. (13), the ties can still occur as we work in hourly resolution. Hence, it is possible that two patients die at the same time. When an event occurs for two patients at the same time, the partial likelihood Eq. (13) is not valid anymore. Several methods exist in the literature to break the ties and remove the ambiguity such as average partial likelihood[35] and *Berslow's method*[36] that lives on two ends of a spectrum. The former takes average among all possible orders of the events that can break the tie. Hence, it is the most accurate method but computationally prohibitive. The latter gives a partial likelihood almost exactly like the original Cox likelihood by assuming that every ordering of tied events results in the same partial likelihood. This method gives a crude estimate but is easy to implement. A midway approach that we adopted in this work is called Efron's tie-breaker[37]. In this method, a weighted average likelihood of tied cases is subtracted from the denominator of Eq. (13). Efron's method gives good accuracy and is moderately easy to work with—see ref. [37] for details.

**Algorithm details**. Survival analysis by the Cox model is done via maximum likelihood estimation, where the aim is to maximise the logarithm of Eq. (10) in the original Cox's proportional hazard model and Eq. (13) in the nonlinear extension. In the original method with linear exponent both gradient $\partial\log\mathcal{L}/\partial\boldsymbol{\beta}$ and Hessian $\partial^2\log\mathcal{L}/\partial^2\boldsymbol{\beta}$ can be computed analytically. This is not the case for our proposed extension Equation (13) with nonlinear exponent. Instead of an analytical gradient, we use automatic differentiation to compute the gradient $\partial\log\mathcal{L}/\partial\boldsymbol{\theta}$. Once the

gradient is derived, an appropriate gradient-based method is used to perturb $\theta$ in the direction that increases the partial likelihood.

As mentioned in "Nonlinear Time-varying Covariates", the linear exponent $\beta^T x$ in Eq. (10) is replaced with a nonlinear function $\phi(\cdot; \theta)$. We use a neural network with $L$ hidden layers to realise this function. The employed network linearly transforms the input features to a $N_{\dim}$-dimensional hidden layer. The transformed features are passed through a leaky rectified linear unit (LeakyReLU)[38] nonlinear activation function. The hidden activations are then transformed by a linear transformation to a single node and finally passes through a tangent hyperbolic (tanh) activation function. In summary the network function can be represented as

$$\phi(\mathbf{x}; \theta) = \tanh(W_2(\text{LeakyReLU}(W_1 \mathbf{x}))), \quad (14)$$

where $\theta = \{W_1, W_2\}$ and $W_i$, $i = 1, 2$ are the trainable weight matrices of the network (Supplementary Fig. 3). We used Xavier's method[39] to initialise the weights $\theta$ of the model. To prevent overfitting, we additionally applied dropout with a dropout probability of $p_{\text{dropout}}$. In our PyTorch[40] implementation of CovEWS, we observed stable convergence of our model using the Adam[41] optimiser with a learning rate of 0.001 for up to 100 epochs.

**Hyperparameter optimisation.** For the methods trained on the Optum training fold (CovEWS and CovEWS [linear]), we used a systematic approach to hyperparameter optimisation where each prediction algorithm was given a maximum of 15 hyperparameter optimisation runs with different hyperparameter configurations chosen at random without duplicates from predefined ranges (see Supplementary Table 5). Out of the models generated in the hyperparameter optimisation runs, we then selected the model that achieved the highest specificity at greater than 95% sensitivity on the validation set of the Optum cohort.

**Postprocessing and calibration.** After training CovEWS using the Optum training cohort, the predicted hazard for a patient with state $\mathbf{x}$ is the hazard function $h(t|\mathbf{x})$ Eq. (9) evaluated at $t = 128$ h ($\approx 5.33$ days) given the current patient covariates $\mathbf{x}$ and under the assumption that patient covariates stay constant. To produce CovEWS scores, we additionally apply post-processing using a percentile transformation that converts $h(t|\mathbf{x})$ into the percentile of patient states in the Optum validation set that are assigned a lower $h(t|\mathbf{x})$ than the evaluated patient state $\mathbf{x}$. We chose to output CovEWS scores in form of percentiles to aid in the clinical interpretation of CovEWS as a risk score relative to a representative set of reference states, and to discourage interpretation as a mortality probability. Interpretation of CovEWS scores as a mortality probability is difficult since the mortality risk of a patient depends on their uncertain future trajectory and the prediction horizon, and is influenced by clinical interventions that may be initiated in the future. As shown in Supplementary Fig. 1, patients' states may change rapidly and frequently, and clinical interventions can significantly and positively alter the trajectory of patients. We also verified experimentally that, when interpreted as a probability of mortality, CovEWS scores overestimate the actually observed probability of death on the Optum and TriNetX test sets since patients' states may improve, due to intervention or otherwise, between the prediction time and the end of the prediction horizon (Supplementary Fig. 4; similar results with CovEWS [linear] Supplementary Fig. 5). We, therefore, decided to instead output CovEWS scores as relative risk percentiles between 0 and 100 that discourages interpretation as a probability of mortality. To aid in the use of CovEWS, the following "Thresholds" outlines calibrated thresholds that can be used to maximise specificity at the desired target level of sensitivity for different prediction horizons.

**Thresholds.** A key question for clinical decision making is which threshold should be used for CovEWS scores to indicate severe risk, and potentially trigger an automated warning. To provide guidance in choosing the appropriate CovEWS score depending on the desired trade-off between sensitivity and specificity, we evaluated the optimal observed thresholds of CovEWS scores for various target sensitivity levels using their respective receiver operator characteristic (ROC) curves for each prediction horizon (Supplementary Table 6). Optimal score thresholds to maximise specificity at high levels of sensitivity were between 61 and 36, 44 and 27, and 34 and 19 depending on the prediction horizon for target sensitivity levels greater than 85%, 90% and 95%, respectively. We note that lower thresholds are necessary to achieve high sensitivity for prediction horizons farther in the future as patients' deterioration has to be identified earlier in its progression.

**Feature importance.** Highlighting the clinical risk factors that positively or negatively influenced CovEWS to output a certain score is of high utility as it enables clinical users to contextualise CovEWS scores, and, in some cases, these highlights could potentially even point towards opportunities for timely intervention. We utilised the differentiability of our prediction model as outlined in "Algorithm Details" to provide a real-time visualisation of the clinical covariates that are most important for CovEWS at any given time point (see Supplementary Fig. 1 for an example). To compute the importance scores at each time point, we used the Integrated Gradients (IG)[42] method that calculates relative importance scores $a_i \in (-100\%, 100\%)$ for each input feature $x_{t,i}$ in the feature vector $\mathbf{x}_t$ with $i \in [0 .. d - 1]$ where $d$ is the number of input features. We used IG with the mean feature vector $\bar{\mathbf{x}}_t$ across the Optum training set as a reference, calculated 50 intermediate steps for

each explained $\mathbf{x}_t$, and normalised $a_i$ to the range of $(-100\%, 100\%)$ by dividing each $a_i$ by the sum $\Sigma_{i=0}^{d-1} |a_i|$ of all feature attributions for $\mathbf{x}_t$. To obtain a timeline of attributions as shown in Supplementary Fig. 1, we calculate attributions $a_i$ whenever a change in patient status was recorded in the patient's EHR.

**Baselines.** In our analysis, we compared the performance of CovEWS to the following existing generic and COVID-19 specific clinical risk scores, and baselines:

*CovEWS (linear).* A linear time-varying survival Cox model as described in "Time-varying Covariates" trained using the same Optum training set and using the same pipeline as the non-linear CovEWS. We used the implementation provided in version 0.24.8 of the lifelines[20] Python package.

*COVID-19 estimated risk for fatality (COVER_F).* The COVER_F scoring system for COVID-19 as described in[19]. Since COVER_F uses a single flag for any heart disease diagnosis, we aggregated all diagnoses in the diagnosis categories ischaemic heart disease, pulmonary embolism, other heart diseases in our dataset into one single joint diagnosis code if any diagnosis in those three categories was present. All other input features used by COVER_F were direct matches with the input covariates of the same name also used by CovEWS (Supplementary Table 1).

*Sequential organ failure assessment (SOFA).* SOFA scoring is commonly used in clinical contexts to indicate the risk of organ failure in critical patients. We, therefore, used SOFA as a generic risk scoring baseline that was not specifically designed for COVID-19 to demonstrate the comparative benefits in the predictive performance of a COVID-19 specific risk scoring system. Since we did not have FiO$_2$ values available in our EHR datasets, we assumed a default FiO$_2$ value of 21% (equal to the fraction of oxygen in inhaled air) for patients that were not intubated, and an average of 71% for patients that are intubated (FiO$_2$ is often set to 100% initially and then progressively lowered as the patient stabilises, see e.g., ref. 43 for an example). In addition, we did not have access to Glasgow coma scale (GCS) scores in the EHRs, and potential additional points from a high GCS score (a maximum of +4) were therefore not reflected in our calculated SOFA scores.

*Modified early warning score (MEWS).* The MEWS[23] is a risk scoring system for patient deterioration used at bedside. Like SOFA, MEWS was not specifically designed for COVID-19, and is therefore a generic risk scoring baseline. Since AVPU (Alert, reacting to Vocal stimuli, reacting to Pain, Unconscious) scores were not available in the EHRs, we assumed a default AVPU state of alert to calculate MEWS scores.

Yan et al. 2020.[17] derived a simple and interpretable decision rule using three features for mortality prediction in COVID-19 patients from data collected from 485 COVID-19 positive patients seen in Wuhan, China. In their validation cohort, the decision rule showed a respectable cross-validated prediction performance of $96.1 \pm 0.03$ (mean ± standard deviation, 5-fold cross validation)[17]. All input features used by Yan et al.[17] were direct matches with the input covariates of the same name also used by CovEWS (Supplementary Table 1).

Liang et al. 2020.[18] developed a prediction model for critical COVID-19 related illness using data from 1590 patients seen at 575 medical centres in China using deep learning and 10 input covariates, including observed X-ray abnormalities. On three external cohorts from different Chinese provinces, they reported a predictive performance in terms of concordance index (c-index) of 0.890, 0.852 and 0.967 for predicting critical illness under the missingness of input covariates, respectively. Since we did not have access to radiologic assessments in our EHR datasets, we evaluated their model with the X-ray abnormality covariate missing for all evaluated patients (i.e., set to zero). To the best of our knowledge, Liang et al.[18] did not specify which co-morbidities were included in their collected dataset. However, their study reports a maximum of 6 co-morbidities diagnosed in one patient. In our evaluation, we counted existing patient diagnoses of pulmonary embolism, kidney disease, inflammatory bowel disease, asthma, rheumatoid arthritis, and diabetes towards these 6 co-morbidities.

**Performance evaluation.** In addition to the results presented in the main body of this work, we also present Receiver Operator Characteristic (ROC) curves for CovEWS for various prediction horizons between 1 and 192 h evaluated on the held-out Optum test set (Supplementary Fig. 7) and the external TriNetX test set (Supplementary Fig. 8), the same ROC curves for CovEWS (linear) (Supplementary Fig. 9 and Supplementary Fig. 10) a comparison of CovEWS, Time Varying Cox[20], COVER_F[19], SOFA[14], MEWS[23], Yan et al.[17], and Liang et al.[18] at various prediction horizons in terms of AUC, AUPR, F$_1$, sensitivity, specificity and specificity at greater than 95% sensitivity (Spec.@95%Sens.) for predicting COVID-19 related mortality on the held-out Optum test set (Supplementary Data 1), on the external TriNetX test set (Supplementary Data 2), on the Optum Future cohort (Supplementary Data 3), and on the Black or African American (Supplementary Data 4), Hispanic (Supplementary Data 5), Asian (Supplementary Data 6), Caucasian (Supplementary Data 7), non-hospitalised (Supplementary Data 8), Fibrin D-dimer (Supplementary Data 9), hsCRP (Supplementary Data 10), Gamma Glutamyl Transferase (Supplementary Data 11), IL-6 (Supplementary Data 12), less than 6 missing covariates (Supplementary Data 13) and less than 9 missing (Supplementary Data 14) subgroups of the Optum test set.

**Software.** The source code used for developing CovEWS and for conducting the presented experiments and analyses was implemented using Python (version 3.7),

scikit-learn (version 0.22.2), numpy (version 1.19.1), scipy (version 1.4.1), pandas (version 1.5.0), PyTorch (version 1.5.1) and lifelines (version 0.24.8). Plots shown were generated using the ggpot2 R package (version 3.3.1), the survival R package (version 3.1-12) and survminer (version 0.4.7).

**Hardware**. We used the high-performance computing (HPC) infrastructure provided by the Personalised Healthcare Informatics group at F. Hoffmann-La Roche Ltd to run the presented experiments. The compute nodes used 1st and 2nd generation Intel Xeon Platinum 8000 series processors and had access to 72 GB random access memory (RAM) each.

## Data availability
The data that support the findings of this study, the TriNetX COVID-19 research and Optum de-identified COVID-19 electronic health record databases, are available from TriNetX, LLC and Optum, Inc. (https://www.optum.com/) but third-party restrictions apply to the availability of these data. The data were used under license for this study with restrictions that do not allow for the data to be redistributed or made publicly available. However, for accredited researchers, the TriNetX COVID-19 research and Optum de-identified COVID-19 electronic health record databases are available for licensing at TriNetX, LLC and Optum, Inc., respectively. Data access may require a data sharing agreement and may incur data access fees.

## Code availability
The code for CovEWS is available at https://github.com/d909b/CovEWS under the MIT license https://opensource.org/licenses/MIT[45].

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

## Acknowledgements

S.P. is supported by the Swiss National Science Foundation under P2BSP2_184359. L.A.C. is funded by the National Institute of Health through NIBIB R01 EB017205. BS is a member of the excellence cluster "Machine Learning in the Sciences" funded by the Deutsche Forschungsgemeinschaft (DFG, German Research Foundation) under Germany's Excellence Strategy-EXC number 2064/1-Project number 390727645. We thank Annika Buchholz for helpful discussions.

## Author contributions

A.M. and P.S. created the new software used in the work. P.S., A.M., S.P., L.A.C., J.H., M.H., S.B. were involved in the conception and design of the work, and the analysis and interpretation of results. J.H., L.A.C., and M.H. provided the clinical motivation and interpretation. All authors were involved in reviewing, drafting and/or editing of the manuscript. P.S. and S.B. supervised the work.

## Competing interests

P.S. is an employee and shareholder of F. Hoffmann-La Roche Ltd. The remaining authors declare no competing interests.
