## [Peer Review File · Nature Communications]

REVIEWER COMMENTS

Reviewer #1 (Remarks to the Author):

Many thanks for the opportunity to review this interesting manuscript on the use of the electronic health records to provide real-time predictions of COVID-19 related mortality. The CovEWS clinical risk scoring system shows a good specificity of up to 78.8%, with a sensitivity greater than 95%. Given the current pandemic, this work can be of value in order to streamline and prioritize care where it is most needed. Also, the CovEWS score can update in real-time without manual action, and accounts for a much wider range of risk factors than a traditional early warning system. This reviewer has several comments.

Major recommendations

(1) The patient cohorts/databases used should be better described in detail. E.g. how was the diagnosis Covid-19 made in these databases (SARS-CoV-2 PCR proven? diagnostic criteria?). Also include details on composition, quality of data used in manuscript, access etc.

(2) A major limitation is the fact that by far the most patients are included before treatment protocols were changed around the world to exclude antibiotics and include remdesivir, dexamethasone and anticoagulant therapy. Of course, with a wide variation in clinical practice. It is thought that current overall mortality is less when compared to situations of a first wave/first responders in this pandemic. The authors address this in a limited fashion by having a secondary analysis's in patients that presented in July when compared to earlier timepoint. At the very least this should be highlighted in the limitation section.

(3) The paper describes the development, validation and performance of the CovEWS tool, but gives limited attention to the practical aspects and limitations of this tool. There is just a few words on the potential implementation of this tool from the clinical perspective. Some thoughts: The main use of such a screening tool is to alert physicians of a patients deterioration, so that they can act upon this. It could help if you would be even more specific on what should be done with an alert by the screening tool. The prediction of mortality itself will not change the outcome, it should be coupled with clear interventions. As an extreme example, you would not start intubating a patient, because the tool says that there is a higher chance of mortality. But what can we do? And at what points during admission should we look at the prediction (e.g. how should it be incorporated in the workflow)? In line, the pitfalls of predicting mortality are being addressed in the discussion. This reviewer would encourage the authors to expand this section, since it is vital information in order to

judge the clinical usability of the tool. Also, numerous algorithms or decision support tools are being developed for all kinds of diseases, and now especially for COVID-19. One of the problems one sees in the literature is that many groups develop such algorithms, but few of them are actually implemented in clinical practice. Can you elaborate on how you plan to introduce this tool into clinical practice and what specific steps are needed to prove the benefits of the tool?

(4) Another use of screening tool development is the discovery of new predictors for certain outcomes. Could the authors elaborate a bit more on the most influential variables in the model, that are associated with mortality? This is very valuable information that could be presented more clearly.

(5) In the baseline characteristics tables, there does not seem to be data on DNR status, which could be an essential predictor of mortality. Was this data available and used?

(6) A related point is the considerable amount of missing data that is being imputed for this model. It is mentioned that missing data is handled with MICE. Can you explain in more detail how the imputation of large amounts of missing data could have affected the performance?

Minor

(7) Is the layout of the manuscript in Nat Commun style?

(8) Could you elaborate on why you chose to present the mortality risk in a 0-100 percentile range? This reviewer believes that it would be hard to translate such a number into a clinical action. An absolute mortality risk could be much clearer.

(9) In the paragraph on “predictive performance for different subgroups”, the more pronounced performance difference for Caucasian and African American populations is discussed. What is the reason for this?

(10) In the results section, an excellent overview of the performance of the CovEWS in relation to other predictors of mortality is given in figure 1. However, the comparison with the SOFA score has some limitations. The SOFA score is based on an intensive care population and should thus do much

better in patient A than in patient B. This reviewer feels that a comparison with the MEWS would be more insightful. The limitation of the SOFA score should at least be acknowledged in this regard.

Reviewer #2 (Remarks to the Author):

This study by Schwab et al. describes the construction, training and validation of a real-time prediction early warning framework (named CovEWS) aimed at predicting the mortality risk of COVID-19 patients. The training of the neural Cox model was based on 66 430 patients from 69 healthcare institutions. The performance of the proposed method was tested using two external cohorts (Optum and TriNetX). Five other baseline approaches were used for comparison purposes and to evaluate the predictive performance of CovEWS. The prediction sensitivities of different methods were provided at 1, 2, 4, 8, 16, 24, 96, 192 hours prior to the observed outcomes. A type of score was also proposed to indicate the real-time importance of input factors. The performance of CovEWS outperformed the baseline methods on both the Optum and TriNetX test datasets and different subgroups. There are several technical and presentation issues to address as follows:

1. Results of the model show that the performance on the Asian subgroup is extremely high than other ethnicities (Figure 2 and Table S13). However, the source of the database used in this study came from US, Australia, Malaysia and India and Asian patients only count for 3.41% in the training set. The authors should further clarify this point.
2. Another concern about this study (and probably the reason for the much better result of the proposed network) is the comparison method. Baseline methods from Yan et al. and Liang et al. show promising results in their studies. However, these studies were done at the initial breakout of COVID-19 with fewer patient samples and most of the patients were Asian. Without re-training or re-tuning these models with datasets used in this study, it is unfair to say the model proposed in this study is superior to these methods.
3. The CovEWS model itself is not interpretable since the authors applied neural networks to enhance feature learning ability of the model. To address this issue, the developed CovEWS model identified clinically-relevant factors for mortality prediction using the Integrated Gradient method (IG). Importance of different factors can be ranked based on the scores produced by IG. However, it seems that this approach cannot provide detailed threshold for these factors in order to set up clinical warning in practice.

4. Since the CovEWS model selected relatively important

factors for mortality prediction, it could be interesting to see whether the CovEWS model is still able to offer accurate prediction using these factors only, which is a nice way to verify the feature selection method.

5. The CovEWS demanded high dimensional inputs for mortality prediction so it is not surprising to see that the model produces accurate prediction. However, in practice, patients may not have complete records, resulting in a large number of missing factors. Although the authors applied MICE to deal with missing values, the percentage of total factors that missing values account for is unknown. It is expected that the prediction accuracy will drop significantly if a large number of factors are missing.

6. When splitting the dataset, the authors wrote: "To demonstrate the generalizability of predictions made by CovEWS, we limited the training of CovEWS to a training cohort of 14 215 (30%) patients from the Optum cohort, used 9 477 (20%) Optum patients for model selection, and evaluated CovEWS against both a held-out test cohort of 14 215 (30%) patients from the Optum cohort." What about the rest 20% patients?

7. It was unclear how to make fair comparisons when there exist a huge number of missing variables. For instance, there are more than 99% values of hs-CRP (one of key variables in [17]) missing in the external validation dataset.

8. TriNex dataset was selected for the external validation in this study. A significant concern about this study is the generalizability. The model performance on this external dataset seems not optimal (even the author explained the missing value problem). Especially, results in the time-varying plot in Figure 3 show that only about 50% of the identified high-risk patients (CovEWS score is 90-100 indicating a very high mortality risk) eventually died. The authors should consider if TriNex is an appropriate dataset for generalizability analysis and explicitly discuss the impact of missing variables.

Minor:

1. Page 25, "Unknown"

2. Figure S4: There are three models after validation. Are they the potential models before hyperparameter optimization?

3. Table S2: $W(1)$ and $W(2)$ should be W_1 and W_2 (as stated in Eq. (14))?

4. Figures 2, 3 and S5: Color contrast of bars/lines/distributions should be increased to highlight the results.

Reviewer #3 (Remarks to the Author):

General comments

=====

I appreciate the opportunity to review this manuscript. By using Optum and TriNetX data, authors developed COVID-19 Early Warning System (CovEWS), which is a clinical risk scoring system for assessing COVID-19 related mortality risk. This article is very well written, and the methods are appropriate. While the predictive performance is acceptable, the clinical impact of this model may be limited.

Specific comments

=====

1. The coding accuracy of the COVID-19 diagnosis is questionable. Are there any validation study using the claims data?
2. What does “COVID-19 positive via a positive lab test” mean? Do the databases include the results of PCR?
3. The use of ICD-9/10 codes as a predictor is not clinically useful. Physicians never code their diagnosis—most of diagnoses are coded after admission or discharge. How does the clinician use the model?
4. The definition of outcome (death) is not validated. In addition to the non-validated study population (ie. COVID-19 diagnosis), the uncertainty in death may bias the results.
5. In the real clinical setting, when a patient come to the ED/outpatient, how “normalize” the patient data?
6. The lack of prospective evaluation limits the generalizability/transportability. In general, the lack of generalizability/transportability is more problematic when using claims data.

7. Predicting high mortality is not equal to predicting optimal timing of treatment. This prediction model never tell us “when we should treat/manage patient.”

8. Lastly, how can we implement the prediction model to the current healthcare system? I understand that the use of machine learning is promising, but the implementation is one of the biggest issues of machine learning. How can we obtain the time-varying data? Any prediction model using machine learning does not have any meanings unless implemented.

To the esteemed Editor and Reviewers,

Thank you for your time, and for providing insightful and constructive feedback on our manuscript. We are pleased that the scope of the manuscript was found to be of interest and the innovation and impact of the work is well recognised by all reviewers. We have put in great effort to satisfactorily address all comments, feedback and input provided, and we have duly incorporated the combined feedback into a revised and improved version of the manuscript.

Please find below our detailed point-by-point responses to the comments received. The abbreviations R1, R2, R3 refer to reviewers 1, 2, and 3 in order of the provided reviews, respectively.

For your convenience, we marked all descriptions of changes we implemented in the revised manuscript with a blue bar - like the one next to this line - on the left margin of this document.

R1: "The patient cohorts/databases used should be better described in detail. E.g. how was the diagnosis Covid-19 made in these databases (SARS-CoV-2 PCR proven? diagnostic criteria?). Also include details on composition, quality of data used in manuscript, access etc."

A.1. We thank R1 (and also R3 in a related question) for these important comments.

Patients were included on the basis of the presence of either (1) a positive recorded SARS-CoV-2 (PCR/RNA) test (the vast majority of patients), (2) the respective COVID-19 diagnostic ICD codes, or (3) both in their electronic health records (EHRs; see also section S.2 "Data Collection", paragraph Inclusion Criteria). In the Optum cohort, which was collected partially during the early phases of the pandemic when testing was still not always available, only a relatively small percentage of 6.31% of COVID-19 diagnoses were not substantiated by a recorded positive SARS-CoV-2 test result (see Fig. S3). Later on in the pandemic, as demonstrated by the Optum Future cohort, all COVID-19 diagnoses were verified by at least one positive SARS-CoV-2 test result in the patient's EHRs (see Figure S3). No significant bias appears to have been introduced by the small fraction of diagnoses without a recorded positive SARS-CoV-2 test result, since the performance of CovEWS remained robust in the Optum Future cohort in which all patients' diagnoses were verified by a positive test result. This finding is in line with a sensitivity analysis performed in a recent epidemiological study using the Optum Research Database that also concluded that there were no significant differences in observed outcomes between COVID-19 patients that were included based on their recorded diagnoses and those included based on positive SARS-CoV-2 tests [1].

[1] Rizzo, Shemra, et al. "Descriptive epidemiology of 16,780 hospitalized COVID-19 patients in the United States." medRxiv (2020).

For further details around data sources, patient inclusion criteria, data quality, included features and other information related to data collection, please see section S.2 "Data Collection" of the manuscript. Data access and licensing is independently managed by the respective federated networks of healthcare organisations, Optum and TriNetX, themselves, and is available to any accredited user (see section 5 "Data Availability").

We note that our analysis was based on de-identified EHRs that include data on demographics, clinical measurements, vital signs, lab tests (including SARS-CoV-2 PCR/RNA tests) and diagnoses. While EHRs are in some jurisdictions also partially used for billing, EHRs contain rich and high resolution information in support of patient care beyond that typically available in claims data.

We added a statement on how COVID-19 diagnoses of included patients were confirmed and a more prominent reference to the further details on data collection presented in section S.2. "Data Collection" of the supplementary material to the main body of the revised manuscript. We also added a reference to [1] to the supplementary material to further reinforce that no epidemiological differences in outcome were observed in patients included without a positive SARS-CoV-2 test result recorded in their EHRs.

R1: "A major limitation is the fact that (...) treatment protocols were changed (...). It is thought that current overall mortality is less (...). The authors address this in a limited fashion by having a secondary analysis's in patients that presented in July when compared to earlier time-point. At the very least this should be highlighted in the limitation section."

R1 is completely correct that changes in treatment guidelines have significantly improved our ability to treat COVID-19 patients. As noted by R1, we evaluated CovEWS on a cohort separated in time (the Optum Future cohort) that was recruited after the RECOVERY trial results (dexamethasone, hydroxychloroquine) had been reported in order to verify that CovEWS remains robust to such changes in treatment regime. The results demonstrated that the predictive performance of CovEWS was sustained in this future cohort (see e.g. Figure 2), despite the significant changes in treatment guidelines. The robustness of CovEWS to changes in treatment guidelines and outcomes stems from its underlying design choices. The momentary risk assessments of CovEWS are based *only* on the current state of the patient, and information about any future treatments that may be applied is not available to the model. Mechanistically, better treatment options help improve patient outcomes by either avoiding or quickly improving critical patient states, and an improved patient state through better treatment will immediately be reflected by CovEWS by outputting a lower risk. Finally, because CovEWS outputs the risk percentile of a patient, it is robust to changes in mortality probability across the range of patient percentiles, i.e. a new treatment guideline may shift the 8-day mortality probability of a 90-percentile patient from 25% to 24% and that of a 80-percentile patient from 15% to 14%. However, since outcomes improved across the range of patient strata, their relative risk ranking remains the same, i.e. a 90-percentile patient is still in the 90th percentile - thereby ensuring CovEWS remains accurate.

We added additional reasoning for the robustness of CovEWS to changes in treatment pattern over time to the "Predictive Performance for Different Subgroups" paragraph in the revised manuscript.

R1: "The paper (...) gives limited attention to the practical aspects and limitations of this tool. (...) Can you elaborate on how you plan to introduce this tool into clinical practice and what specific steps are needed to prove the benefits of the tool?"

A.2. We thank R1 (and R3) for this extremely important consideration. Indeed, as correctly highlighted by R1 (and also R3 in another question), the clinical implementation and translation of systems like CovEWS remains an important step to fully realising their value to patients. As noted by R1, there are several potential use cases for CovEWS each requiring different implementation considerations.

There are two potentially attractive options to implement CovEWS in clinical practice: (1) a standalone web application with a front-end for entering clinical risk factors, and (2) integration with existing electronic medical records (EMRs) systems, such as Epic and Cerner. To deploy CovEWS as a standalone web application, a server hosting the model and providing a user interface to enter and upload the required patient information in a privacy-preserving HIPA-compliant manner would have to be provided. A standalone web application is a flexible solution because it does not require an EHR system to be in place, nor a deep integration with any particular EHR system - which can in many cases be fairly complex to implement. Any user with an internet connection could access this web application and request predictions for their patient data. However, a solution based on a standalone web application is not the most convenient solution from a user perspective, as data would have to be entered manually to obtain predictions. In such a scenario, one would likely restrict their usage to request predictions only at specific pivotal timepoints along the patient journey, such as for example the admission of a patient or after a significant change in patient status.

In contrast, for an implementation based on a deep integration with an EHR system, the primary components would be (i) a data ingestion module that automatically reads data from a patient's EHR and processes it for input into CovEWS, (ii) a visualisation component that displays the patient's CovEWS score and the importance of the most relevant clinical risk factors to medical staff (see Figure 1 for an example), and (iii), optionally, an alert module that automatically generates alerts based on elevated CovEWS scores and transmits them to subscribing medical staff. Compared to the standalone solution, the deeply integrated solution requires more custom integration effort and collaboration with EHR vendors. However, it also offers more convenient automated information processing and visualisation from a user perspective.

Since both solutions are valuable to different user groups, we are currently pursuing both potential solutions. We are exploring the provision of a standalone web application dashboard on a pro bono basis that serves CovEWS as part of its functionality. In parallel, we are looking for potential first partner HCOs that would work with us to deeply integrate CovEWS into their EHR systems. With those HCOs, as mentioned by R1, we would have to define entry points to integrate with their existing clinical workflows. Potential integrations into clinical workflows are highly dependent on the existing HCO workflows and their context, and therefore have to be defined individually for each partner HCO. Interesting pivotal timepoints along the patient journey include pre-testing, admission, discharge, prior to and after significant interventions such as intubations, and when monitored in critical care on a continuous basis.

Potential future steps for validating CovEWS further include conducting prospective studies that evaluate whether the provision of CovEWS improves resource use and patient outcomes by enabling more accurate care prioritisation and more timely intervention. For example, a randomised controlled trial (RCT) comparing key outcomes (mortality, length of stay, critical care admissions) between groups of patients for which (i) only standard early warning systems or (ii) standard early warning systems plus CovEWS are available could be conducted to prospectively evaluate CovEWS. In addition, CovEWS could be studied further to elucidate whether certain combinations of patient states, treatments and CovEWS scores are associated with better or worse outcomes - which, if successful, could potentially in the future provide intervention guidelines on top of CovEWS.

We added additional discussion on the potential integration points of CovEWS into clinical workflows to section 4 "Discussion" of the revised manuscript.

R1: "Could the authors elaborate a bit more on the most influential variables in the model, that are associated with mortality? This is very valuable information that could be presented more clearly."

We thank R1 for this insightful recommendation. A number of known risk factors, including age, SpO₂, blood pressure, ischemic and other heart diseases, hypertension, white blood cell count, neutrophils, lymphocytes, high-sensitivity C-reactive protein, creatinine, lactate dehydrogenase, albumin, pH and PCO₂, were found to be significantly ($p < 0.05$) associated with mortality outcomes in the Optum dataset (Table S4). We note that, as correctly pointed out by R1, these factors are merely associated with mortality and do not imply causation.

We added a list of the variables that were found to be significantly associated with mortality outcomes, and a reference to Table S4 that shows the observed p-values of association with mortality outcomes for all input variables to section 4 "Discussion" of the revised manuscript.

R1: "In the baseline characteristics tables, there does not seem to be data on DNR status, which could be an essential predictor of mortality. Was this data available and used?"

No, do not resuscitate (DNR) status was unfortunately not available in our datasets.

We clarified that DNR status was not available in our datasets, and that DNR status may have sig-

nificant impact on observed patient mortality in sections 4 "Discussion" and S.2 "Data Collection" of the revised manuscript.

R1: "Can you explain in more detail how the imputation of large amounts of missing data could have affected the performance?"

R2: "It is expected that the prediction accuracy will drop significantly if a large number of factors are missing."

As correctly noted by R1 and R2, we used MICE to impute missing data for patients for which not all data was observed in the dataset. MICE has recently been independently reported as the recommended approach to address missingness in early warning score systems by Gerry et al [2]. Large differences between the imputed values for a missing entry and the true underlying value that was not measured may lead to prediction errors larger than those that would have been achieved if the missing entry was available. However, since missingness is inherent in real-world data and requiring all data to be present is therefore impractical, designing systems to be able to use all available information and imputing any information that is missing via MICE is the preferred approach to handling missing data.

The impact of higher missingness can for example be seen when comparing the non-hospitalised and the Optum Test cohorts, since the former has considerably higher missingness (see Table S6). Quantitatively, the higher missingness in the non-hospitalised subgroup slightly impacted the prediction performance of CovEWS primarily on the shorter prediction horizons (see Tables S9 and S16). For longer prediction horizons, no noticeable impact of the higher missingness was discernible (Tables S9 and S16) - likely because of the generally higher uncertainty over longer prediction horizons.

[2] Gerry et al. Early warning scores for detecting deterioration in adult hospital patients: Systematic review and critical appraisal of methodology. BMJ 2020

We added a reference to [2] to section S.2 "Data Collection" of the revised manuscript.

R1: "Could you elaborate on why you chose to present the mortality risk in a 0-100 percentile range? This reviewer believes that it would be hard to translate such a number into a clinical action. An absolute mortality risk could be much clearer."

We chose to output CovEWS scores in form of percentiles to aid in the clinical interpretation of CovEWS as a risk score relative to a representative set of reference patient states, and to explicitly and actively discourage direct interpretation as a mortality probability (see also section S.9 "Postprocessing and Calibration" of the manuscript).

We chose to discourage direct interpretation of CovEWS scores as a mortality probability for two reasons (S.9 "Postprocessing and Calibration"):

Firstly, CovEWS is a momentary risk assessment based only on the current state of a patient that, in particular, does not have access to information about interventions that may potentially be planned by medical staff in the future. If CovEWS were to output mortality probabilities, a very high mortality probability, such as for example 98%, could easily be misread by clinical staff as interventions being futile - when in reality the high risk output would be based only on the *current* critical state of the patient which may be improved significantly with the right actions in the future. In contrast, as a percentile score, CovEWS scores would read as: "In her current state, the patient has a mortality risk higher than 98 percent of observed patient states in the reference cohort." This alternative reading as a percentile risk score avoids any potentially misleading interpretation by medical staff beyond a momentary assessment, and makes clear that future actions can significantly alter the patient's risk of mortality.

Secondly, any mortality probability output by CovEWS could only be calibrated for one specific prediction horizon, for example for the 8 hours prediction horizon, and would therefore be

incorrect for all others. Depending on the context, different prediction horizons may have higher clinical utility. The reading as a percentile score as outlined above is correct *independent of the prediction horizon* and may therefore be used more flexibly by medical staff using the optimal thresholds as appropriate for the patients individual situation and desired prediction horizon (Table S8).

Finally, as outlined above, treatment guidelines and protocols and available treatment options may change significantly in the future. When interpreted as a risk percentile, CovEWS remains more robust to such changes than a system based on assigning mortality probabilities that may have to be re-trained and re-distributed frequently to account for such changes - which would cause considerable logistical challenges.

R1: "In the paragraph on "predictive performance for different subgroups", the more pronounced performance difference for Caucasian and African American populations is discussed. What is the reason for this?"

The sentence was meant to highlight the more pronounced predictive performance difference between CovEWS and other risk assessment methods in the Caucasian and African American subgroups compared to the Asian subgroup - which may be reflective of the fact that several baselines have been developed using data from predominantly Asian populations.

We adapted the wording of the sentence in question to further clarify its intention.

R1: "The SOFA score is based on an intensive care population and should thus do much better in patient A than in patient B. This reviewer feels that a comparison with the MEWS would be more insightful."

We thank R1 for this excellent suggestion.

We added the Modified Early Warning Score (MEWS) as an additional comparison to Figures 1 and 3 of the revised manuscript, and to the results tables in the Supplementary Material.

R2: "Results of the model show that the performance on the Asian subgroup is extremely high than other ethics (Figure 2 and Table S13). (...) The authors should further clarify this point."

Most methods, with the exception of MEWS, appear to perform better on the Asian subgroup compared to the held-out Optum test cohort. Since the Asian subgroup is small compared to the other cohorts (only 513 patients; Table 2), it is possible that the Asian subgroup was systematically different to the held-out Optum test cohort, and not representative for all Asian patients. We would therefore not attribute this difference to the amount of training data available for Asian people for CovEWS, but rather to underlying differences in the populations.

We further highlighted the limitation that several subgroups are relatively small and therefore likely not representative of all patients included in that group in section S.2. of the revised manuscript.

R2: "Baseline methods from Yan et al. and Liang et al. (...) were done at the initial break-out of COVID-19 with fewer patient samples and most of the patients were Asian. Without re-training or re-tuning these models with datasets used in this study, it is unfair to say the model proposed in this study is superior to these methods."

We thank R2 for this important comment. We believe it is important and fair to independently evaluate the performance of the models published by Yan et al. 2020 and Liang et al. 2020 in an external test cohort precisely because their cohorts were small and non-representative. For example, the quantitative results presented in our manuscript indicate that the machine-derived

decision rule presented in Yan et al. 2020 does not apply in our larger, more representative cohort - which is a noteworthy result in itself. In addition, our manuscript presents a number of modelling innovations, including a novel time-varying formulation for neural Cox models, an associated training procedure, and an associated explanation method based on Integrated Gradients (IG). Moreover, neither of the compared methods, including CovEWS, were retrained on the external TriNetX cohort, and the superior predictive performance of CovEWS was maintained.

R2: "(The) importance of different factors can be ranked based on the scores produced by IG. However, it seems that this approach cannot provide detailed threshold for these factors in order to set up clinical warning in practice."

In practice, warnings should be set up based on thresholds defined based on the patient's CovEWS scores themselves (S.10 "Thresholds"). Our answer above to the question raised by R1 and R3 around potential translation of CovEWS outlines how such an alert-generating system could be implemented in clinical practice.

We also note that, while simplified decision rules may be easier to apply in clinical practice than a machine-learning system without any electronic infrastructure, they also perform significantly worse in terms of predictive performance as demonstrated by our experimental evaluation (see Figure 2; Yan et al. 2020, MEWS, SOFA and COVER_F are such simplified decision rules).

R2: "Since the CovEWS model selected relatively important factors for mortality prediction, it could be interesting to see whether the CovEWS model is still able to offer accurate prediction using these factors only, which is a nice way to verify the feature selection method."

We thank R2 for highlighting this important point. The feature importance scores produced alongside CovEWS predictions using the integrated gradient (IG) method update over time as the patient's status changes (see Figure 1). It is therefore not possible to use these feature importance scores for feature selection, since the ranking of most important features changes depending on the patient's status. Features used by CovEWS were selected for inclusion based on existing evidence and expert input (see S.2 "Data Collection" - paragraph "Feature Selection").

As outlined in our response to the question on influential variables raised by R1 above, we have further highlighted the variables that we found to be most significantly associated with mortality outcomes (Table S4) in the revised manuscript.

R2: "When splitting the dataset, the authors wrote: (...) What about the rest 20% patients?"

We thank R2 for catching this error. The training cohort indeed corresponded to 23 692 patients (50%) of the Optum data as shown in Table 1, not 30%.

We adapted sections 3 "Results" and S4 "Stratification" in the revised manuscript to show the correct absolute and relative size of the training fold of the Optum data.

R2: "It was unclear how to make fair comparisons when there exist a huge number of missing variables. For instance, there are more than 99% values of hs-CRP (one of key variables in [17]) missing in the external validation dataset."

We thank R2 for this important consideration. All covariates are imputed via MICE prior to evaluating the models (see S.5 "Preprocessing"). All methods were therefore fairly evaluated on the same patients even if some of their key variables were missing prior to imputation. We note that hs-CRP in particular can be imputed accurately to a range of roughly 3 mg/l using CRP (which was more commonly sampled at a missingness rate of 55% in TriNetX). While [17] used hs-CRP for their model, the cutoff used is at 41.2 mg/l, and the performance of the Yan et al. 2020

baseline would therefore only be affected negatively when the observed value is very close to the cutoff when utilising the imputed information from CRP.

Finally, we believe the chosen evaluation setting is fair to all methods: Firstly, because if the information for a covariate is not available in a patient, then no method can use it to gain an advantage over other methods, and, secondly, because (2) methods that are reliant on covariates that are rarely available in practice have less clinical utility. Our comparison is fair, because it gives a realistic estimate of what predictive performance could be expected under constraints to data collection as observed in a large and representative patient sample. In contrast, evaluating a method only on patients where the highest quality of information is always available would not be fair and give a misleading picture of the performance that could be expected in clinical practice, since this quality of information would not be available in all cases in real world settings.

R2: "TriNetX dataset was selected for the external validation in this study. (...) The model performance on this external dataset seems not optimal (even the author explained the missing value problem). (...) The authors should consider if TriNetX is an appropriate dataset for generalizability analysis and explicitly discuss the impact of missing variables."

We thank R2 for this important feedback. While the predictive performance of CovEWS was indeed slightly lower on TriNetX than on Optum (Figure 2), it was still significantly ($p < 0.05$) superior to all other risk scoring systems - in many cases by a considerable margin.

The key question when assessing generalisability of CovEWS is whether CovEWS maintains a high degree of predictive performance in new healthcare organisations, contexts and environments for which it did not have training data. Given that TriNetX differs in geography and population (TriNetX includes international hospitals), data collection (section S.2 "Data Collection"), and missingness patterns (Tables 1 and 2 and Figure S5) and covers a large number of patients (5 005), it is a good choice of external validation data set precisely because it tests the robustness of CovEWS to variations in all these factors. Conversely, a dataset that is similar in all these factors to the original training set would potentially demonstrate a higher performance of CovEWS but it would also be much less informative as to the robustness of CovEWS to variations in missingness, data collection and geography and environment, and therefore make for a poor external validation set. We therefore believe the heterogeneous characteristics of TriNetX make it a good external test set.

Please see our answer above to the question raised by R1 about missingness for a discussion of the effect of missingness on model performance.

R2: "Page 25, "Unknown""

We thank R2 for this suggestion.

In the revised manuscript, we changed the sentence structure to avoid the use of the quoted "Unknown".

R2: "Figure S4: There are three models after validation. Are they the potential models before hyperparameter optimization?"

Yes, the interpretation by R2 is correct. Please note that the number of candidate models shown in Figure S4 is illustrative. As described in section S8 "Hyperparameter Optimisation", 15 candidate models were available for selection after the hyperparameter optimisation.

R2: "Figure S6: $W(1)$ and $W(2)$ should be W_1 and W_2 (as stated in Eq. (14))?"

We thank R2 for catching this oversight.

We changed the notation of W(1) and W(2) in Figure S6 to match the notation used in Eq. (14).

R2: "Figures 2, 3 and S5: Color contrast of bars/lines/distributions should be increased to highlight the results."

We thank R2 for this excellent suggestion.

We changed the colors in Figures 2, 3 and S5 of the revised manuscript.

R3: "The coding accuracy of the COVID-19 diagnosis is questionable. Are there any validation study using the claims data?"

R3: "What does "COVID-19 positive via a positive lab test" mean? Do the databases include the results of PCR?"

Please see our combined answer A.1 provided to similar set of questions posed by R1 above.

R3: "The use of ICD-9/10 codes as a predictor is not clinically useful. Physicians never code their diagnosis—most of diagnoses are coded after admission or discharge. How does the clinician use the model?"

We thank R3 for this important comment. In our experimental evaluation, information from diagnostic codes is only used from the time point it was entered into the patient EHR, i.e. the predictive performance shown is already reflective of real-world coding behaviours as observed across more than 69 HCOs. In our datasets, the majority of 85.0% and 72.5% of all diagnoses observed in the TriNetX and Optum cohorts, respectively, were coded in the EHR and available *before* inclusion of a patient into the cohort based on a COVID-19 diagnosis or positive SARS-CoV-2 test result. The remainder of diagnoses was added after the COVID-19 diagnosis of the patient. A timely entry of relevant EHR data is in many cases mandated by institutional guidelines as these data points may be needed to guide and organise patient care, and our experimental results show that the data as recorded across the 69 HCOs enables CovEWS to operate accurately.

We added the above details on the timing and availability of diagnoses in our evaluated cohorts to section S.2 "Data Collection" of the revised manuscript.

R3: "The definition of outcome (death) is not validated. In addition to the non-validated study population (ie. COVID-19 diagnosis), the uncertainty in death may bias the results."

We thank R3 for this important consideration. Mortality outcomes used in our analysis were validated at source by the healthcare organisations (HCOs) that contributed their de-identified EHR data. In the US context, HCOs' death records would typically be based on a death certificate. We note that HCOs can only include mortality outcomes in their records when they become aware of them, and that this is most likely to occur if a patient dies while under their care. For outpatients, there may be a systematic bias stemming from the fact that not all patients that die outside of HCO care would be recorded, because the HCO may not in all cases be aware of these death events. However, the impact of this bias on CovEWS is likely limited, since we focus on predicting mortality outcomes that happen between 1 and 192 hours in the future based on the patient's current state, and patients that died within 1 to 192 hours after being seen were likely still under the care of the HCO. In contrast, patients that died as outpatients most likely had an unknown state between 1 and 192 hours prior to their death, and including their deaths in the training of CovEWS would therefore in any case not be informative.

We added a statement on the limitation of potentially incomplete death records particularly for outpatients in section 4 "Discussion" of the revised manuscript.

R3: "In the real clinical setting, when a patient come to the ED/outpatient, how "normalize" the patient data?"

Depending on the chosen implementation of a solution based on CovEWS (see answer above regarding the clinical translation of CovEWS), patient data normalisation would either be handled (1) at the user interface layer by providing exact input instructions and formats in a standalone web application, or (2) at the data ingestion layer in a solution that is embedded with an EHR system. In both an embedded system and a standalone application, the normalisation can be performed automatically and seamlessly by software based on the underlying data model (see section S.3 "Data Normalisation").

R3: "The lack of prospective evaluation limits the generalizability/transportability."

We evaluated CovEWS on the external TriNetX cohort consisting of data from 5 005 patients from 24 HCOs with different geography and population (TriNetX includes international hospitals; Table 1), data collection (section S.2 "Data Collection"), and missingness patterns (Tables 1 and 2 and Figure S5). The evaluation demonstrated that CovEWS maintained its high predictive performance relative to other risk assessment systems even when applied to new hospitals with different environments, missingness patterns, and underlying populations.

In addition, we evaluated CovEWS on a third cohort separated in time that was collected after the RECOVERY trial results were reported and after CovEWS was developed. The experimental results showed that CovEWS maintained its high predictive performance relative to other risk assessment systems even under changing treatment guidelines and other temporal effects. Moreover, we performed a subgroup analysis that indicated the performance of CovEWS is sustained across various ethnic groups and cohorts.

We believe these results provide strong evidence that the predictive performance of CovEWS generalises well to new environments, and is robust to changes in treatment and data collection policies and environmental conditions.

R3: "Predicting high mortality is not equal to predicting optimal timing of treatment. This prediction model never tell us "when we should treat/manage patient."

CovEWS is a risk assessment system that may be used to prioritise care and enable early intervention. We do not claim that CovEWS can be used to predict the optimal timing of treatment. However, systems like CovEWS can have significant clinical utility in practice since assessing a patient's individual risk accurately is often difficult, particularly in complex systemic diseases, such as COVID-19, and when many clinical factors are involved. As demonstrated by our experimental results, CovEWS has, in most cases, a significantly better predictive performance than existing generic and COVID-19-specific risk scoring systems across all analysed prediction horizons and cohorts, and may therefore enable critical time for the initiation of interventions and prioritisation of scarce healthcare resources for those most in need. Moreover, CovEWS can be used as an indicator of treatment success, since a patient that does not improve in CovEWS score over time after a treatment has been initiated may potentially require further intervention to manage their mortality risk. Nonetheless, CovEWS is primarily designed to be an additional, more accurate information source for medical staff that can without further studies inform, but not guide, their decision making.

R3: "How can we implement the prediction model to the current healthcare system? I understand that the use of machine learning is promising, but the implementation is one of the biggest issues of machine learning. How can we obtain the time-varying data? (...)"

Please see our combined answer A.2 provided to a similar question posed by R1 above.

We hope that our revised and improved manuscript satisfactorily addresses all comments, feedback and input provided by the reviewers, and very much look forward to hearing back from you soon.

Kind regards,

Dr Patrick Schwab
for the authors.

REVIEWER COMMENTS

Reviewer #1 (Remarks to the Author):

I would like to thank the authors for their comprehensive answers and incorporation of the feedback.

This reviewer feels that the questions have been adequately addressed and that the manuscript has improved substantially. Especially the clinical aspects have been highlighted more clearly.

Reviewer #2 (Remarks to the Author):

Real-time Prediction of COVID-19 related Mortality using Electronic Health Records

I appreciated the efforts that the authors put in improving the quality of their manuscript. Yet, my major concerns have not been well addressed.

1. First and foremost, I was not convinced by the performance and comparison results using datasets that has more than 99.28% missing measurements for one of the key variables used in different models. In other words, there is only 1 in every 140 patients that has a measurement. How can one infer/estimate the measurements of other patients based on this single measurement?

2. MICE is good for imputing missing values in normally distributed and categorical variables and has been applied to many applications [ref]. But there is a more fundamental problem before applying MICE: what is the largest percentage that MICE can impute with a trustable performance? When the ratio of missing values exceeds this threshold, the imputed values become random guess. Then the conclusion drawn could be wrong!

[ref] Gerry et al. Early warning scores for detecting deterioration in adult hospital patients: Systematic review and critical appraisal of methodology. BMJ 2020

3. My previous concerns on the fairness of comparison using largely missing datasets still hold. There are too many missing values for variables in ALL compared models. Using, for instance, values of CRP to estimate those of hs-CRP is not a good idea. Not only they have different means (Table), they have different impact on (Table S4). One shall notice that the p-values were calculated using χ^2 tests are different (0.52 versus < 0.005).

The imputation method (MICE) accompany the discussion of the effect of the missingness (and its corresponding reference) can be moved to the main text, as all three Reviewers raised concerns about the impact of this issue.

4. I appreciated the effort that puts together a cohort of 66,430 COVID-19 positive patients seen at over 69 healthcare institutions in the United States (US), Australia, Malaysia and India. Yet, it would make the paper much more convincing if the authors can have a small yet (almost) complete dataset at least for comparison.

All in all, I have no issue that the authors proposed a risk score, yet I have concerns about their reported performance, given the issues listed above.

Reviewer #3 (Remarks to the Author):

The author have answered my questions appropriately.

To the esteemed Editor and Reviewers,

Thank you for your insightful feedback on the provided response and the revised manuscript. We are pleased that the majority of the provided responses to the comments raised in the first review round were found to be satisfactory, and we are happy to address the remaining concerns herein. We have again duly incorporated the most recent feedback in a revised and improved version of the manuscript.

Please find below our detailed point-by-point responses to the latest comments received. The abbreviations R1, R2 and R3 respectively refer to reviewers 1, 2 and 3 in order of the originally provided reviews.

For your convenience, we marked all descriptions of changes we implemented in the revised manuscript with a blue bar - like the one next to this line - on the left margin of this document.

R1: "I would like to thank the authors for their comprehensive answers and incorporation of the feedback. This reviewers feels that the questions have been adequately addressed and that the manuscript has improved substantially. Especially the clinical aspects have been highlighted more clearly.."

We thank R1 for their insightful comments and feedback that have helped improve our manuscript substantially.

R2: "I was not convinced by the performance and comparison results using datasets that has more than 99.28% missing measurements for one of the key variables used in different models. (...) How can one infer/estimate the measurements of other patients based on this single measurement?"

We thank R2 for this relevant follow-up question. We believe that the question raised by R2 points to two misunderstandings that we would like to clarify further.

Firstly, the 99.28% missingness cited by R2 refers to the missingness reported for the hsCRP measurement in the external TriNetX test set (Table 1). The missingness observed in hsCRP in the external TriNetX test set is not relevant for the estimation performance of MICE because TriNetX was an external and completely independent dataset, and therefore *not* used to derive the MICE estimators. Instead, the MICE estimators were derived from the training set, and thereafter applied to all other folds and datasets (see Section S.5 "Preprocessing"). The training set consisted of 23 692 patients with a substantially lower missingness of 96.62% in hsCRP (Table 1) - which corresponds to a respectable number of more than 800 patients with one or more hsCRP measurements available for MICE to learn the relationship between hsCRP and the other covariates. With 3.38% of patients available for training, the number of patients available is - even in the most extreme case of hsCRP - considerably higher than the 0.72% cited by R2.

Secondly, we note that the missingness number cited by R2 is also not relevant for the estimation performance of hsCRP values in TriNetX, because the hsCRP values are estimated by MICE using the *other* covariates. The missingness ratios relevant for the estimation performance of MICE for hsCRP are therefore all the other measurements' missingness ratios and the strengths of their respective associations with hsCRP. CRP in particular is strongly associated with hsCRP (see following responses), and available frequently in patients (Table 1). The 99.28% missingness cited by R2 only influences (i) how often imputation is performed for hsCRP, and (ii) how often hsCRP is available to MICE for the imputation of other measurements in the external TriNetX test set.

In the revised manuscript, we further highlighted that the relations used to impute covariate values were derived from the Optum training set, and that a respectable number of at least 800 ($\approx 3.38\%$) patients were available for this purpose for all measurements, including hsCRP¹

¹We note that the training set missingness of hsCRP is the highest for any measurement included in our study

R2: "(...) there is a more fundamental problem before applying MICE: what is the largest percentage that MICE can impute with a trustable performance? When the ratio of missing values exceeds this threshold, the imputed values become random guess. Then the conclusion drawn could be wrong!"

A.1. We thank R2 for the important consideration. As outlined in the previous question, the percentage of missing values is not the relevant quantity for determining the expected estimation performance of MICE, and there is therefore no a priori threshold at which MICE can not be used [1]. Instead, what matters is how much information about the estimated covariate can be derived from all the other covariates that are available at a given time point. This can be demonstrated with a simple thought experiment: Suppose we are measuring two covariates $x_1 \sim \mathcal{N}(0, 1)$ and x_2 where $x_2 = 2x_1$. The (simple) relationship between x_1 and x_2 can be learnt even if the training data has a very high missingness in x_2 as long as enough paired samples are available for the MICE estimator in the training set, and x_2 could, once learnt using data from the training set, accurately be estimated in a target dataset even if x_2 was not measured *at all* in the target dataset (i.e. 100% missingness²). We note that the estimates produced by MICE are never merely a random guess, but always the best estimate given all the available information for that patient at that time point based on associations derived from the training set.

Nonetheless, R2 is absolutely correct that (i) there may be differences between the best estimate produced by MICE and the true unmeasured value of a covariate, and that (ii) such differences may lead to prediction errors. Since missingness is common in clinical practice, we aimed to develop and experimentally evaluate a clinical risk prediction system for COVID-19 related mortality that can be used in the presence of missingness in our study. Given that the use of prediction systems will generally involve missing entries in practice, we believe it is *required* to evaluate their performance under the presence of realistic missingness patterns - as we have done in our experimental evaluation. As mentioned previously in our response to the prior comments raised by the reviewers, our chosen approach, MICE, is the widely accepted standard and recommended approach to handling missingness in this setting [2].

In contrast, the alternative proposed by R2, to evaluate prediction systems only on complete cases without considering any missingness, ignores the fact that missingness is both common and characteristic in clinical practice. As outlined in our prior response, an evaluation only on complete cases would not be a realistic representation of the performance that would be encountered in real-world care settings, where missingness is pervasive and must be expected as not all tests and observations are available at all times for all patients. In addition, a complete case analysis would be biased, as patients for which more tests have been performed are significantly different to those for which not all information was collected (see Tables S9 and S10 in the revised manuscript). For example, patients for which more information was collected may have had longer stays during which there was more opportunity for testing, or they may more likely have been particularly complex cases where more tests were necessary to establish the underlying condition (Section S.15 in the revised manuscript).

We therefore believe our presented analysis is more meaningful, robust and representative for the envisioned use case than the proposed alternative complete case analysis.

[1] Medley-Dowd et al. The proportion of missing data should not be used to guide decisions on multiple imputation. *Journal of Clinical Epidemiology* 2019

[2] Gerry et al. Early warning scores for detecting deterioration in adult hospital patients: Systematic review and critical appraisal of methodology. *BMJ* 2020

(tied with gamma glutamyl transferase [GGT], see Table 1), and MICE therefore in all cases had 800 ($\approx 3.38\%$) or more patients available to learn the association between the target measurement and the other measurements.

²This thought experiment also further exemplifies why the missingness of hsCRP in TriNetX as mentioned in the previous question is not relevant for the estimation performance of MICE - even at 100% missingness in TriNetX the value of hsCRP may still be estimated accurately if the relationship between hsCRP and the other covariates has been learnt from the training data.

In the revised manuscript, we further highlighted that our experimental evaluation sought to compare methods for COVID-19 related mortality risk prediction in real-world conditions, which include among others missingness, variations in treatment policies and differences in treated populations at different sites, as observed in a large and representative cohort across multiple hospitals. We also added the limitation that certain methods may have originally been developed and evaluated only on complete cases to Section 4 "Discussion".

R2: "My previous concerns on the fairness of comparison using largely missing datasets still hold. There are too many missing values for variables in ALL compared models. "

We very much appreciate this concern raised by R2. However, we respectfully, but firmly, disagree with the notion that the presented evaluation is not fair³. Overall, there are four main approaches to comparing different risk prediction systems under missingness. Out of the four main options, we believe that the option we have chosen is the fairest and most representative for the envisioned clinical use.

Option 1: Evaluate on all patients, impute missing information (our chosen approach).

Using Option 1, all prediction systems are evaluated on all COVID-19 patients, irrespective of whether any of their covariates may have originally been missing. Missing covariates are imputed using MICE - the widely accepted and recommended standard approach to handling missingness in this setting - and the prediction systems therefore receive a full set of covariates for all patients. As outlined previously, there may be differences between the true underlying value of a covariate and the covariate value estimated by MICE. However, since all methods receive input covariates imputed in the same manner, they are all impacted in the same way by potential imputation errors - making Option 1 fair to all prediction systems, since no system can gain an undue advantage over any other as neither system has access to the true underlying values.

Option 2: Evaluate on all patients, do not predict when required covariates not available.

Option 2 corresponds to a clinical use pattern in which a prediction system is only used when all required covariates are available (if the system is not designed to handle missing information). Using Option 2, the predictive performance in terms of sensitivity of systems not designed to handle missingness would be strictly lower than the one calculated using Option 1 since they would not be able to detect mortality events for patients for which required covariates are missing. In the case of Yan et al. 2020, this would translate to a maximum possible sensitivity close to 0%, even if all mortality events in complete cases were correctly identified, because hsCRP is rarely available. We therefore believe Option 2 is not as fair as Option 1 because it does not give all system equal opportunity to identify mortality events in the same patients.

Option 3: Evaluate only on patients with all data available. As also outlined previously, evaluating only on complete cases may misstate the predictive performance of risk prediction systems by evaluating them only on a non-representative subset of patients. Moreover, the relevant subset of patients (each with potentially their own unique sampling bias) is different for each prediction system, since they have different required covariates. Option 3 makes comparisons of multiple prediction systems in a representative manner impossible, since the union of required covariates for all compared systems reduces the number of leftover patients down to very small, non-representative cohorts that would likely be strongly affected by sampling bias, even in a very large original cohort such as the one we evaluated against. In terms of fairness, Option 3 is also problematic because predicting mortality for the subset of patients that have all required covariates available may be easier or harder than for the overall population of COVID-19 patients - therefore giving a misleading picture of the predictive performance that could be expected in a realistic care setting.

Option 4: Do not compare performance at all on datasets with missingness. Option 4

³Based on the questions raised by R2 in the previous review round, we herein assume R2 argues that the presented evaluation is not fair to other risk prediction systems, i.e. that other prediction systems are disproportionately disadvantaged by the chosen evaluation approach.

would imply that the performance of prediction systems can not be compared in realistic settings, since missingness is characteristic for real-world care settings (Table 1). Given that the real-world performance of prediction systems is of paramount scientific and practical interest, we believe that Option 4 is not a valid option.

Among the main evaluation options, we believe our chosen approach is therefore both the fairest and also the most representative for the presented clinical setting (see also our response A.1).

Finally, we performed further analyses on six additional sub-cohorts with fewer missing covariates in response to a suggestion raised by R2 in a following question (A.2). The experimental results of the additional analyses demonstrate that the use of missingness imputation did not disproportionately disadvantage any specific prediction system, and therefore did not preclude the fair evaluation of prediction systems in our study.

We added the above discussion to Section S.16 "Evaluating Risk Predictors under Missingness" to the supplementary material of the revised manuscript, and a reference to the various considerations when evaluating risk prediction systems under missingness to Section 5 "Discussion" of the revised manuscript.

R2: "Using, for instance, values of CRP to estimate those of hs-CRP is not a good idea. Not only they have different means (Table), they have different impact on (Table S4). One shall notice that the p-values were calculated using χ^2 tests are different."

High-sensitivity C-Reactive Protein (hsCRP; <https://loinc.org/30522-7/>) and C-Reactive Protein (CRP; <https://loinc.org/1988-5/>) tests both quantify the serum level of CRP (albeit with different sensitivity), and it is therefore sensible to use the latter to inform the imputation of the former. Any differences in medians shown in Table 1 are due to differences in for whom and when the tests were ordered by clinical staff. Clinically, hsCRP is generally used to estimate low-grade inflammation, and will therefore more often be ordered to measure lower values of CRP - which is why the median values of hsCRP shown in Table 1 are slightly lower than those of CRP. If measured at the same time in the same patients, the result of a hsCRP test would be expected to be highly correlated with a CRP test - which is the ideal setting for missingness imputation using MICE (see the thought example presented in our previous response A.1 above). We note that we included both tests in our datasets despite their high correlation since they are used interchangeably in different situations by clinical staff.

The referenced χ^2 tests in Table S4 assess the goodness-of-fit of the CovEWS (linear) model parameters that were trained on the already-imputed dataset, i.e. the p-values in Table S4 can not be used to reason about the relationship between hsCRP and CRP prior to imputation.

R2: "The imputation method (MICE) accompany the discussion of the effect of the missingness (and its corresponding reference) can be moved to the main text, as all three Reviewers raised concerns about the impact of this issue."

We thank R2 for the excellent suggestion.

We added a discussion on missingness and the relevant citation to Section 5 "Discussion" of the revised manuscript.

R2: "I appreciated the effort that puts together a cohort of 66,430 COVID-19 positive patients seen at over 69 healthcare institutions in the United States (US), Australia, Malaysia and India. Yet, it would make the paper much more convincing if the authors can have a small yet (almost) complete dataset at least for comparison."

A.2. We thank R2 for the suggestion to add a comparison of the evaluated prediction systems on a smaller, more complete sub-cohort. We added the requested further analyses (using

six additional subcohorts with fewer missing covariates) to the revised manuscript. While extrapolation beyond these specific subcohorts is difficult due to their non-representative nature (see our responses provided above), the results of the analysis demonstrated that the use of missingness imputation did not disproportionately disadvantage any included prediction system since the overall ranking of prediction systems and their relative predictive performance differences were, with few exceptions, sustained in all the subcohorts with fewer missing covariates - where missingness imputation was either used more sparingly or not at all for certain covariates (Figure S9 and Tables S19 to S24 of the revised manuscript).

In the revised manuscript, we added Section S.15 "Subcohort Analysis with Fewer Missing Covariates" that presents further analysis of six additional subcohorts consisting of patients (i) for which the four covariates with the highest missingness ratios (Fibrin D-dimer, hsCRP, Gamma Glutamyl Transferase, IL-6) were available at least once in their EHR (Table S9), and (ii) for which a maximum of respectively 6 and 9 covariates were not available in their EHR (Table S10). We also added a reference to these additional analyses to Section 5 "Discussion" of the revised manuscript.

R2: "All in all, I have no issue that the authors proposed a risk score, yet I have concerns about their reported performance, given the issues listed above."

We appreciate the encouraging feedback, and hope our above responses and the further analyses provided in the revised manuscript give sufficient reassurance that MICE is a suitable methodology to handle missingness in the presented datasets, and that our chosen evaluation approach was both fair and representative for the envisioned clinical use.

R3: "The author have answered my questions appropriately."

We thank R3 for the constructive and insightful feedback, and are pleased that the comments were found to be addressed satisfactorily.

We hope that our revised and improved manuscript satisfactorily addresses all comments, feedback and input provided by the reviewers, and very much look forward to hearing back from you soon.

Kind regards,

Dr Patrick Schwab
for the authors.

REVIEWERS' COMMENTS

Reviewer #2 (Remarks to the Author):

The author have answered my questions appropriately.